# Structure of the Sec14 domain of Kalirin reveals a distinct class of lipid-binding module in RhoGEFs

Yunfeng Li [1], Yulia Pustovalova[1], Tzanko I. Doukov [2], Jeffrey C. Hoch [1], Richard E. Mains [3], Betty A. Eipper [1,3] & Bing Hao [1] ✉

Gated entry of lipophilic ligands into the enclosed hydrophobic pocket in stand-alone Sec14 domain proteins often links lipid metabolism to membrane trafficking. Similar domains occur in multidomain mammalian proteins that activate small GTPases and regulate actin dynamics. The neuronal RhoGEF Kalirin, a central regulator of cytoskeletal dynamics, contains a Sec14 domain (Kal^bSec14) followed by multiple spectrin-like repeats and catalytic domains. Previous studies demonstrated that Kalirin lacking its Sec14 domain fails to maintain cell morphology or dendritic spine length, yet whether and how Kal^bSec14 interacts with lipids remain unknown. Here, we report the structural and biochemical characterization of Kal^bSec14. Kal^bSec14 adopts a closed conformation, sealing off the canonical ligand entry site, and instead employs a surface groove to bind a limited set of lysophospholipids. The low-affinity interactions of Kal^bSec14 with lysolipids are expected to serve as a general model for the regulation of Rho signaling by other Sec14-containing Rho activators.

The Sec14 superfamily is defined by the presence of a CRAL_TRIO domain (Pfam, PF00650), best known for its ability to bind small lipophilic molecules[1–5]. The functions of family members that contain only a Sec14 domain range from coordinating lipid metabolism and vesicular trafficking in yeast to providing 11-cis-retinal to retinal pigment epithelial cells and Mueller glia for delivery to photoreceptor cells[3,6,7]. A third of the ~30 human Sec14 superfamily members also contain enzymatic domains that control activities of the Rho family small GTP-binding proteins[8,9].

Based on their sequences and structural features, Sec14 superfamily members fall into two sub-families, CRAL_TRIO and BCH[1]. Structural, biochemical and functional studies of yeast Sec14p, the prototypical CRAL_TRIO sub-family member, have guided subsequent studies[10–12]. Despite relatively limited sequence identity, members of the CRAL_TRIO sub-family for which structural information is available adopt a characteristic two-lobed globular structure. The CRAL_TRIO motif harbors a large hydrophobic pocket in the center of its α/β fold with a mobile helical gate controlling access of lipophilic ligands to the internal pocket[3,11]. A small N-terminal lobe, referred to as a CRAL_TRIO_N domain (Pfam, PF03765), closes off one end of the lipid-binding pocket with an all-helical tripod-like fold that can interact with bound lipid head groups. The CRAL_TRIO_N domain also participates in subcellular targeting of Sec14 domain proteins[13,14]. The only BCH sub-family member for which structural data are available is *Schizosaccharomyces pombe* p50RhoGAP, in which intertwined asymmetric monomers contribute to a uniquely different lipid/RhoA-binding pocket[15].

The Sec14-like domains of five mammalian Rho GDP/GTP exchange factors (RhoGEFs) share limited sequence identity with both BCH and CRAL_TRIO sub-family members[1,3,8]. SESTD1, a synaptic regulator of Rho GTPases, falls into this sub-group, but lacks a RhoGEF domain[16]. Each of these six proteins lacks a CRAL_TRIO_N domain but contains a putative CRAL_TRIO domain followed by multiple spectrin-like repeats (SRs), a common structural element consisting of a three-helix bundle.

[1]Department of Molecular Biology and Biophysics, University of Connecticut Health Center, Farmington, CT 06030, USA. [2]Macromolecular Crystallography Group, Stanford Synchrotron Radiation Light Source, SLAC National Accelerator Laboratory, Stanford University, Stanford, CA 94309, USA. [3]Department of Neuroscience, University of Connecticut Health Center, Farmington, CT 06030, USA. ✉e-mail: bhao@uchc.edu

The N-terminal CRAL_TRIO domain of Kalirin is followed by nine SRs and a variable number of RhoGEF (consisting of DH and PH domains), SH3 and protein kinase domains[17] (Fig. 1a). The ability of Kalirin to participate in downstream signaling pathways initiated by multiple receptor tyrosine kinases, G-protein coupled receptors, cell/cell adhesion complexes and ligand-gated ion channels identify Kalirin as a signaling hub in the nervous system[17–19], heart[17], bone[20] and skeletal muscle[20]. Mutations in *KALRN* and changes in its expression have been associated with schizophrenia, ischemic stroke, Alzheimer's disease and vascular disease[17–19]. Kal7, the major isoform expressed in the adult brain, supports the formation and function of excitatory synapses[19]. Kal7 lacking its Sec14 domain still stimulates spine formation, but the spines produced are short; consistent with this, expression of the Sec14 domain alone increases spine length but not spine number[21]. The mechanisms through which the Kalirin Sec14 domain interacts with specific membranes, lipids or proteins to affect spine length and synaptic transmission remain elusive.

The tissue-specific and developmentally regulated use of alternate initiation exons (Ex1) places different peptides immediately upstream of the Kalirin CRAL_TRIO domain. Ex1B encodes a hydrophilic negatively charged front peptide, producing bKal7, while Ex1C generates a positively charged front peptide that forms an amphipathic helix, producing cKal7[20]. Attachment of the Ex1C front peptide to the CRAL_TRIO domain of Kalirin enables its direct interaction with liposomes in a phosphoinositide-dependent manner; attaching it to EGFP localizes EGFP to the Golgi region[20]. When preceded by the Ex1B front peptide, the ability of the CRAL_TRIO domain to interact with phosphoinositide-containing liposomes is eliminated[20]. The ability of

these front peptides to alter Sec14 domain-mediated membrane interactions is essential to their function.

In this report, we present the crystal structure of the CRAL_TRIO domain of Kalirin preceded by the Ex1B front peptide (Kal$^{bSec14}$), the only Sec14 structure for a RhoGEF. Kal$^{bSec14}$ alone adopts a Sec14p homologous CRAL_TRIO domain fold, but with a closed conformation impeding access to the canonical ligand entry site. Our biochemical, biophysical and mutagenesis studies demonstrate that the amino acid residues lining a surface groove on Kal$^{bSec14}$ are directly involved in the binding of specific lysophosphatidylcholine (LPC) lipids and that the Ex1B peptide enhances lipid binding affinity. Both secondary-structure predictions and structure-based alignments identify Kal$^{bSec14}$ as the prototype of a Sec14 fold sub-class, with a unique ligand-binding site and ligand specificity (hereafter called SecSR sub-class). Coupled with interactions mediated by their other domains, linear peptide motifs and numerous phosphorylation sites, our findings suggest that the low-affinity interactions of SecSR domains with membrane lipids play an important role in the ability of Kalirin and other SecSR-containing RhoGEFs to participate in rapid regulation of their subcellular localization and signaling functions.

## Results
### Production and structure determination

The CRAL-TRIO domain of Kalirin is sandwiched between alternative start sites in Ex1A, B, C, and D of the *KALRN* gene and the nine sequential SRs that follow its C-terminus (Fig. 1a). To optimize crystallization, we designed multiple CRAL-TRIO-containing constructs of Kalirin and examined their protein expression level as well as

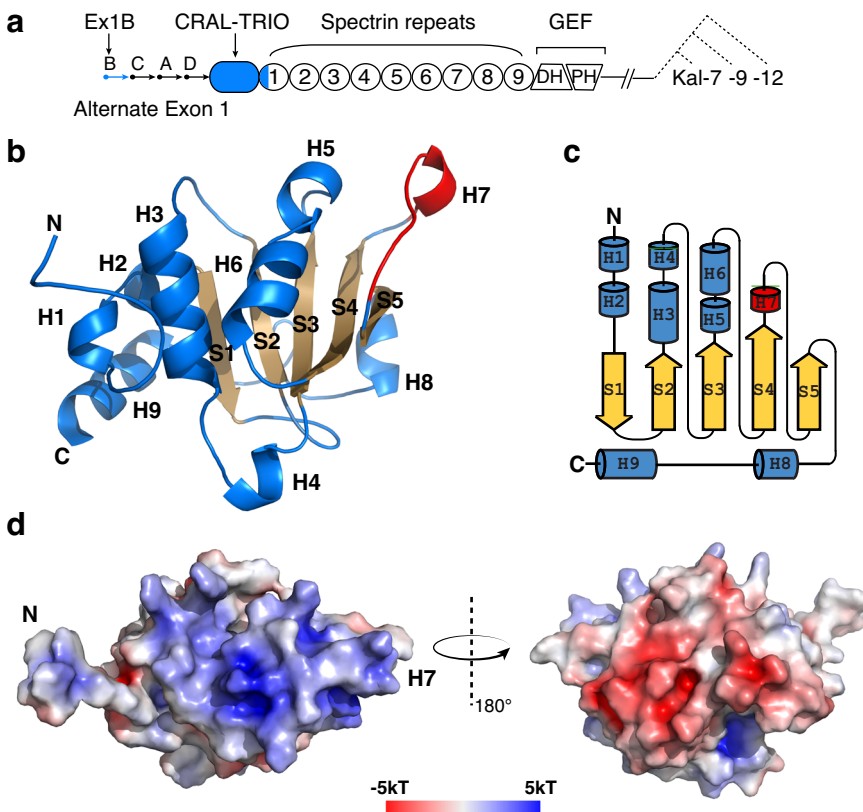

**Fig. 1 | Crystal structure of Kal$^{bSec14}$. a** Schematic diagram of common domains in major isoforms of Kalirin (Kal7, 9 and 12) and use of the four *KALRN* promoters that encode alternate first exons (Ex1A, 1B, 1C and 1D). The CRAL_TRIO domain begins in exon 2, which is common to transcripts initiated at each of these four promoters. The Kal$^{bSec14}$ construct used in this study consists of the Ex1B front peptide, the CRAL_TRIO domain and the A helix of the first SR, all colored in blue. **b** Ribbon diagram of Kal$^{bSec14}$, with secondary structure elements labeled. The helices are shown in blue, the H7 helix in red and the β strands in orange. The first 17 residues of the Ex1B peptide are intrinsically disordered in the structure and the most N-terminal residue shown in this diagram is Val18. **c** Topology diagram of Kal$^{bSec14}$. Cylinders and arrows represent helices and β strands, respectively. **d** Molecular surface representation of Kal$^{bSec14}$ shown in an orientation similar to that in **b** and colored according to the local electrostatic potential calculated with the program ABPS[70].

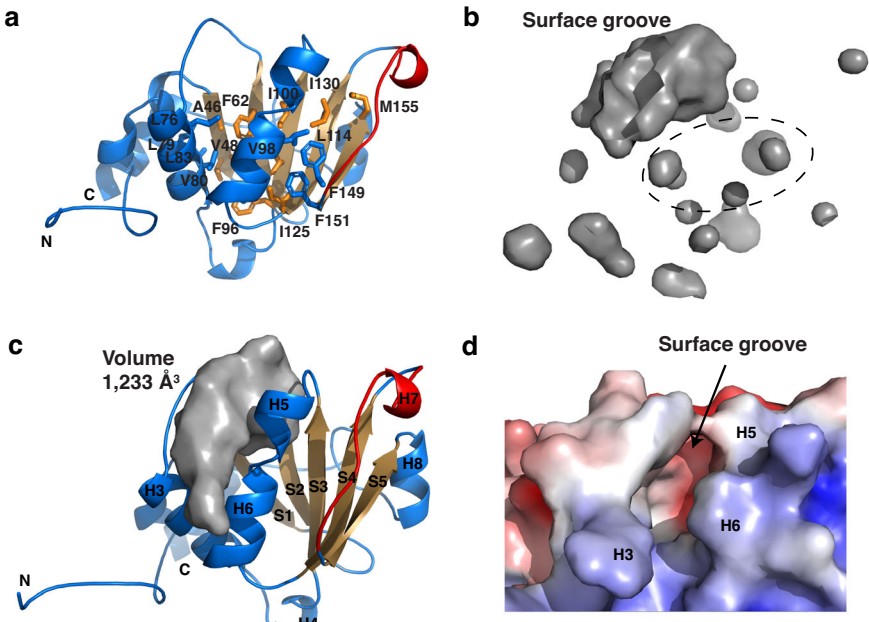

**Fig. 2 | Kal^bSec14 adopts a closed conformation containing a surface groove.**
**a** Residues lining the presumed internal pocket are shown as licorice sticks.
**b** Surface representation of the inner and surface cavities of Kal^bSec14 shown in an orientation similar to that in **a**. The cavity detection radius and cutoff are set at 5 Å and 3 Å, respectively, as defined by Pymol. Dashed ellipse indicates the location of the internal pocket. **c** Representation of the surface groove of Kal^bSec14 generated by HOLLOW[23]. The exterior envelope is set at 8 Å³ over the surface atoms. The volume of the surface groove was calculated with CASTp[24]. **d** Molecular surface representation of the surface groove of Kal^bSec14, colored according to the local electrostatic potential as calculated with ABPS[70].

crystallization propensity. We obtained diffraction-quality crystals of Kal^bSec14, a construct consisting of the Ex1B front peptide (residues 1–25), the putative CRAL-TRIO domain (residues 26–182) and a part of the predicted A-helix (residues 183–192) of the first SR (Fig. 1a). Kal^bSec14 was produced in *Escherichia coli* and the recombinant protein is monomeric as determined by size exclusion chromatography. The X-ray crystal structure of Kal^bSec14 was determined at 1.89-Å resolution by single-wavelength anomalous dispersion using a selenomethionine derivative and refined to an R factor of 19.5% and an $R_{free}$ value of 21.8% (Supplementary Table 1). The asymmetric unit contains eight nearly identical protein monomers, with an overall root mean square deviation (rmsd) of 0.8 Å for 175 $C_\alpha$ atoms (Supplementary Fig. 1). The refined model for monomer B in the PDB file, which is complete except for the intrinsically disordered first 17 residues of the Ex1B front peptide, was chosen for detailed structural analysis.

## Kal^bSec14 structure

Kal^bSec14 adopts an αβα sandwich fold similar to the prototypical CRAL-TRIO domain in yeast Sec14p[10], but with unique features (Fig. 1b, c, and Supplementary Fig. 2a). The central β sheet is composed of five strands (S1–S5), with S1 the only antiparallel strand, and is flanked by six α helices (H1–H6) and a $3_{10}$ helix (H7) on one side and by two α helices, H8 and H9, on the other. An N-terminal 10-amino-acid extension (residues 18–27) upstream of the H1 helix projects ~18 Å away from the αβα core, confirming the notion that the Sec14 domain of Kalirin lacks the CRAL_TRIO_N domain found in many stand-alone Sec14 proteins. Interestingly, Ile186 and Leu190 of helix H9 engage in van der Waals interactions with Ala34, Val37, Leu38, Ile40 and Leu41 of helices H1 and H2. In addition, the bulky side chains of Tyr180, His182 and Trp185 on the H9 helix stack with the benzene ring of Phe47 from strand S1. As such, the interaction of the C-terminal H9 helix, which is part of the first SR, with the central αβα fold buries 441 Å² of solvent-accessible surface area. In the absence of the H9 helix, diffraction-quality crystals of the Kalirin Sec14 domain were not obtained, suggesting that this interaction stabilizes the core fold of the Kalirin CRAL_TRIO domain.

Despite their low degrees of sequence homology, the CRAL-TRIO domains of Kal^bSec14 and several CRAL_TRIO sub-family proteins can be superimposed with a $C_\alpha$ rmsd of <5.0 Å (Supplementary Fig. 2b). For example, Kal^bSec14 and yeast Sec14p[10] share only 16.1% sequence identity but display a 4.0-Å rmsd for 149 equivalent $C_\alpha$ atoms (Supplementary Fig. 2b, c). A comparison of Sec14p paralogs from yeast and orthologs from various fungal species demonstrated conservation of surface charge in a dipole pattern, with conspicuous electropositive regions localized near the ligand binding site and gating helix, thereby facilitating potential interactions of each protein with an electronegative membrane surface[12]. Consistent with this notion, Kal^bSec14 exhibits an asymmetric electrostatic potential distribution on its solvent-exposed surface, with one side largely electropositive and the other electronegative (Fig. 1d). Therefore, the CRAL-TRIO domain of Kal^bSec14 is a structurally conserved element that may govern membrane binding specificity.

## A closed conformation and unique surface groove

Previous crystallographic studies revealed the presence of a phospholipid/detergent-binding pocket with a volume of ~2,000 Å³ in an open yeast Sec14p conformer[10] (Supplementary Fig. 3a). This hydrophobic pocket, a hallmark of Sec14 proteins examined to date, is required for Sec14p to coordinate regulation of lipid metabolism, lipid signaling and vesicular trafficking[11]. Access to this binding pocket is controlled by a helical gate that is flipped open in Sec14 domain apo-structures and closed in ligand-bound structures[11]. Unexpectedly, Kal^bSec14 has only several small, discontinuous hydrophobic pockets (~100 Å³ each) in the center of its αβα fold, none of which is large enough to accommodate even a single phospholipid molecule (Fig. 2a, b). A string of hydrophobic residues from strands S1–S4, helices H3 and H6, as well as the H6–S4 and H7–S5 loop regions line these pockets and render the entire region extremely hydrophobic. Moreover, the $3_{10}$ helix H7, along with an extended loop, is positioned across any potential openings into these internal pockets, mimicking the closed conformation of the so-called mobile gate as often observed for holo-forms of other Sec14 proteins (Supplementary Fig. 3b). Helix

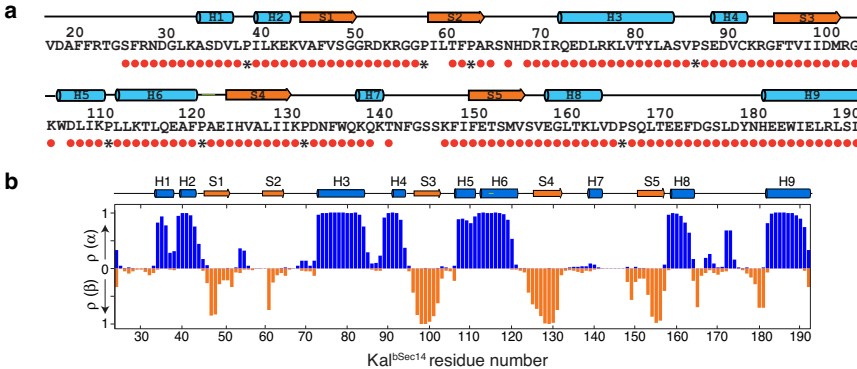

**Fig. 3 | Sequence-specific NMR backbone assignments for Kal^SecI4. a** Sequence of the Kal^bSecI4 residues visible in the crystal structure; red dots signify assigned residues. Proline residues are indicated by black asterisks. **b** Secondary structure propensity of Kal^SecI4 derived using TALOS-N[62]. Probabilities of occurrence for helices [ρ(α)] and β-sheet [ρ(β)] are shown in blue and orange, respectively. The corresponding secondary structure elements from the X-ray structure of Kal^bSecI4 are shown across the top. Source data are provided as a Source Data file.

H7 makes extensive van der Waals contacts with helix H5 and strands S3 and S4, further stabilizing this conformation (Fig. 2a). We therefore conclude that Kal^bSecI4 exists in a closed conformation even in the absence of ligand.

A number of computer programs[22–24] were used to analyze potential ligand-binding sites in Kal^bSecI4; none detected any interior pocket with a volume comparable to the pockets found in other Sec14 family proteins (Fig. 2b). Nevertheless, an elongated groove is present on the surface of Kal^bSecI4, directly above the S2 and S3 strands. The groove has a volume of ~1200 Å³ and is ~20 Å long, 8 Å wide and 8 Å deep (Fig. 2c and Supplementary Fig. 4a). Segments of two strands (Phe62–Ala64 of S2 and Ile100–Met102 of S3) constitute the floor of the groove, while the S2–H3 loop (Arg65–Arg72) together with a segment of the H3 helix (Asn73–Lys78) frames one side of the groove, and the other side is flanked by the S3–H5 loop (Arg103–Ser105) and segments of the H5 and H6 helices (Lys106–Lys111 and Lys113–Leu117, respectively). Interestingly, this groove displays a dipole-like pattern of surface charge distribution, with the S2–H3 and S3–H5 loop regions at one end being largely electronegative and the H3 and H6 helices at the other being electropositive (Fig. 2d).

A comparison of the distribution of internal and exposed pockets of Kal^bSecI4 and other CRAL_TRIO sub-family proteins reveals that canonical CRAL_TRIO domains lack a surface groove-like feature. Each of the six structures determined for CRAL_TRIO sub-family proteins encloses a large internal hydrophobic pocket extending from the gating helix to cover most part of the central β sheet, to which most ligands bind (Supplementary Fig. 4b). None of these CRAL_TRIO domains possess an exposed cavity in a location similar to the Kal^bSecI4 surface groove. In α-TTP complexed with both α-tocopherol (α-Toc) and phosphatidylinositol 4,5-bisphosphate (PIP2), α-Toc is bound deep in its hydrophobic core, while PIP2 is found in close proximity to the gating helix[25] (Supplementary Fig. 4b). Although the binding site for α-Toc is right below the site equivalent to the surface groove in Kal^bSecI4, the α-Toc site is entirely sequestered from bulk solvent. Among the five Sec14 sub-family proteins with a CRAL_TRIO_N domain preceding their CRAL_TRIO domain (Sec14p, CARALBP, Sfh1, Sfh5 and α-TTP), each CRAL_TRIO_N domain is distant from the location of the presumed surface groove but adjacent to the bottom of the gating helix, allowing it to interact with the head group of the bound ligand (Supplementary Fig. 4b).

Expanding on the finding of the Kal^bSecI4 surface groove, we explored its ability to interact with water-soluble phosphatidycholine (PC) mimetics (Supplementary Table 2). These PC mimetics, also called phosphocholine detergents, were selected for the study because PC is the most abundant phospholipid in mammalian membranes[26]. We were unable to obtain cocrystals of Kal^bSecI4–phosphocholine complexes. Their interactions were therefore characterized using solution nuclear magnetic resonance (NMR) spectroscopy, which allows detection and characterization of weak, transient interactions[27].

## Binding of PC mimetics to Kal^bSecI4 via its surface groove

We used NMR chemical shift perturbation (CSP) measurements of Kal^bSecI4 in the absence and presence of PC mimetics to determine ligand-interacting residues and ligand binding affinities. As expected for a well-folded structural unit, well-dispersed signals with adequate intensity were detected in the acquired two-dimensional (2D) ¹H-¹⁵N heteronuclear single quantum coherence (HSQC) spectrum of Kal^bSecI4 in native-like conditions (Supplementary Fig. 5a). 178 out of 187 expected backbone amide resonances were observed. Because NMR chemical shift mapping experiments are only capable of defining protein-ligand interactions when highly complete specific backbone assignments are available, we next sought to determine sequence-specific backbone assignments for Kal^bSecI4. However, a high degree of spectral crowding in the N-terminal disordered region of Kal^bSecI4 (between 8.0 and 8.5 ppm on the ¹H axis) made it difficult to unambiguously identify individual cross-peaks. To simplify NMR data collection and subsequent data analysis, a shorter construct referred to as Kal^SecI4 was generated. Kal^SecI4 (residues 25–192) consists of the CRAL-TRIO domain and the A-helix of the first SR but lacks the Ex1B front peptide. The 2D ¹H-¹⁵N-HSQC spectrum of Kal^SecI4 features an extremely well-dispersed set of resonances (Supplementary Fig. 5b). Importantly, the vast majority of the amide resonance peaks of Kal^bSecI4 and Kal^SecI4 are well matched, facilitating the confident transfer of backbone assignments (Supplementary Fig. 5c).

A total of 92.7% of the backbone ¹H and ¹⁵N resonances of the 160 non-proline residues, and 92% of the ¹³Cα, 91% of the CO and 88% of the expected ¹³Cβ resonances were unambiguously assigned for Kal^SecI4 based on a standard set of triple resonance spectra (Fig. 3a and Supplementary Fig. 5d). The absence of several peaks and the presence of weak peaks prevented a higher percentage of resonance assignments. The backbone amide residues that could not be assigned are localized in two clusters, which include residues in the S2 strand and the S2–H3 loop and in the H7–S5 loop region. Importantly, the secondary-structure elements of Kal^SecI4 predicted based on the backbone and Cβ chemical shifts are in excellent agreement with the secondary structures observed in Kal^bSecI4, as determined by X-ray crystallography, indicating that Kal^SecI4 adopts a fold similar to that of Kal^bSecI4 (Fig. 3b).

We used CSP mapping to probe interactions between ¹⁵N-labeled Kal^bSecI4 and PC mimetics of varying lengths (Supplementary Table 2). Overlays of the ¹H-¹⁵N HSQC spectra of Kal^bSecI4 display gradual shifts in the amide resonance positions for a number of peaks upon titration of increasing amounts of tetradecylphosphocholine (FC14) or

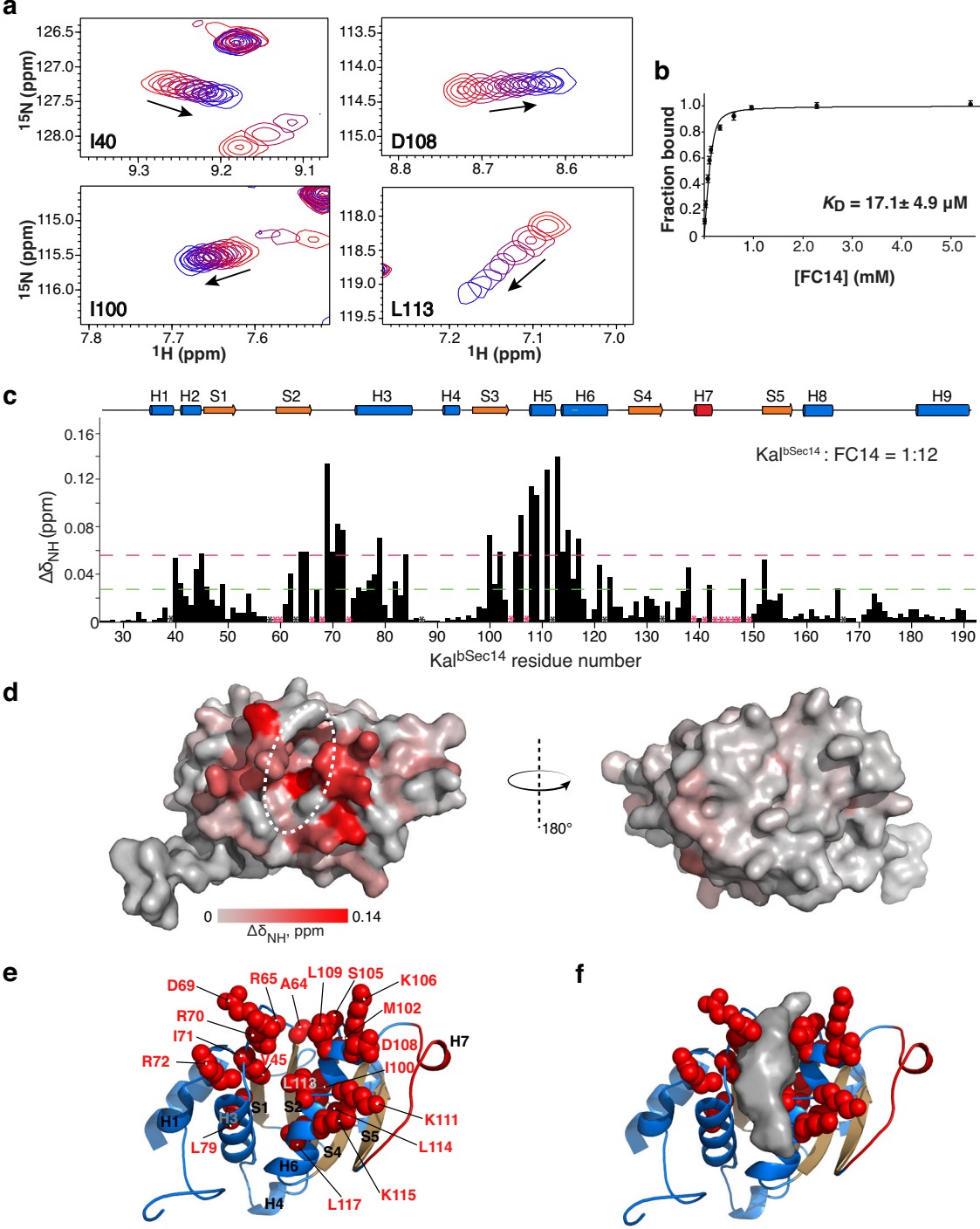

**Fig. 4 | Kal^bSec14 binds FC14 via the surface groove. a** Four residues of Kal^bSec14 that demonstrate CSPs when titrated with increasing amounts of FC14 (1:12 final molar ratio). For each residue, the cross peaks are color-ramped from red to blue with increasing FC14 concentrations, as indicated by arrow. The full series of spectra for the titration are shown in Supplementary Fig. 6a. **b** A plot of normalized global fitting of the averaged CSPs ($\triangle\delta_{obs}/\triangle\delta_{max}$) as a function of FC14 concentration to estimate the $K_D$ for binding as described in Methods. The fitting data and the $K_D$ value represent the mean ± standard deviation (S.D.) of the CSP data for individual residues ($n = 20$). Source data are provided as a Source Data file. **c** Plot of per-residue backbone CSPs between free and FC14-bound states of Kal^bSec14. Proline residues and residues missing backbone assignment are indicated by an asterisk (black, proline; red, unassigned). Dashed green and red lines indicate CSP values within one (1σ) and two (2σ) S.D. of the average chemical shift distribution (0.026 ppm) among all assigned residues, respectively. Source data are provided as a Source Data file. **d** Surface representation of Kal^bSec14 colored according to CSPs induced by FC14 binding, from light gray (no observed CSP) to red (maximum CSP). Dashed ellipse indicates the surface groove. **e** Residues exhibiting the largest FC14-induced CSPs (>2σ) are shown as red spheres; the orientation of Kal^bSec14 matches that shown on the left side of **d**. **f** Surface representation of the surface groove surrounded with the residues specified in **e**.

dodecylphosphocholine (FC12), indicative of specific interactions between Kal^bSec14 and FC14/FC12 and fast exchange between the free and bound states on the NMR timescale (Fig. 4a, b, and Supplementary Fig. 6). In striking contrast, no peak shifts are observed when decylphosphocholine (FC10) or hexadecylphosphocholine (FC16) is added. Per-residue CSPs are calculated as the distance between the free and FC14-bound peaks (Fig. 4c). Of 160 assigned peaks, 19 experience CSPs greater than two standard deviations (2σ) of the

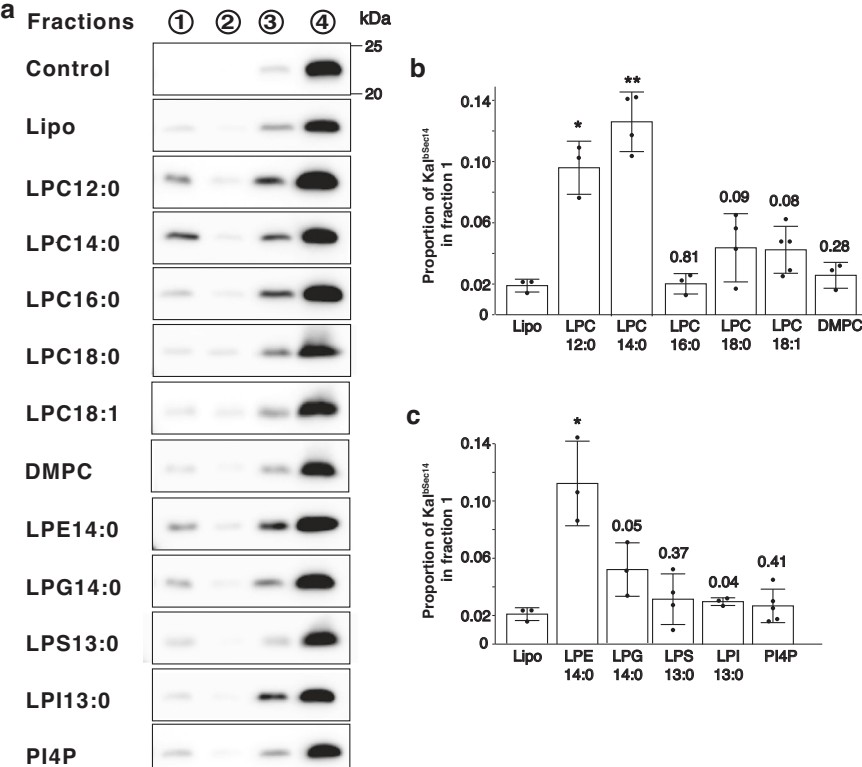

**Fig. 5 | Interaction of Kal$^{bSec14}$ with phospholipids. a** Representative Western blots from flotation assays with Kal$^{bSec14}$ alone (Control), and Kal$^{bSec14}$ with the control lipid mix (Lipo) or with the lipid mix containing the indicated additional lipid. After incubation at room temperature, the Kal$^{bSec14}$/liposome mixtures were placed onto an Accudenz cushion, overlaid with buffer containing lower concentrations of Accudenz and subjected to centrifugation. Four fractions were then collected from each gradient; the top fraction (fraction 1) contains at least 90% of the Liss Rhod PE marker included in the liposome mixture. The protein content of each fraction was quantified by Western blotting using an antibody to the Sec14 domain of Kalirin.

Source data are provided as a Source Data file. **b**, **c** Quantified group data showing protein content of the top fraction for each condition shown in **a**. Stated value is normalized to total protein recovered from each gradient. Graph bars represent the mean ± S.D. of three ($n = 3$ for control, Lipo, LPC12:0, LPC16:0, DMPC, LPE14:0, LPG14:0 and LPI13:0), four ($n = 4$ for LPC14:0, LPC18:0 and LPS13:0) or five ($n = 5$ LPC18:1 and PI4P) independent experiments. Black dots indicate individual data points. *P*-values were determined using two-tailed Student's *t*-test. Statistical significance was determined *versus* Lipo mix as values of $p < 0.02$ (**$p < 0.001$; *$p < 0.02$; listed if considered not statistically significant).

average chemical shift distribution. Binding of FC12 to Kal$^{bSec14}$ exhibits large CSPs to a similar set of residues as those with FC14. The fitting of the NMR titration data obtained with a two-state binding model results in dissociation constants ($K_{Ds}$) of $17.1 ± 4.9\ \mu M$ and $245 ± 36\ \mu M$ for FC14 and FC12, respectively, revealing a 14-fold higher binding affinity for FC14 (Fig. 4b and Supplementary Fig. 6d). In the absence of the Ex1B front peptide, Kal$^{Sec14}$ also exhibits significant CSPs for a similar set of residues (Supplementary Fig. 7). The approximately two-fold attenuation in the affinities of FC14 and FC12 for Kal$^{Sec14}$ ($K_D = 35.6 ± 7.2\ \mu M$ and $514 ± 66\ \mu M$, respectively) suggests a role for the front peptide in ligand binding (Supplementary Fig. 7c, f).

NMR chemical shift mapping of the FC14-induced backbone CSPs onto the Kal$^{bSec14}$ structure reveals that the majority of the residues most sensitive to FC14 binding (CSP > 2σ) are those forming the aforedescribed surface groove (Fig. 4d). These residues form patches that line both sides of the groove: Ala64, Arg65, Asp69, Arg70, Ile71, Arg72 and Leu79 from strand S2, helix H3, and the loop between them are on one side, and Ile100, Met102, Ser105, Lys106, Asp108, Leu109, Lys111, Leu113, Leu114, Lys115 and Leu117 along the entire face of helices H5 and H6 are on the other side (Fig. 4e, f). Similar CSPs are induced in Kal$^{Sec14}$ upon complex formation with FC14 (Supplementary Fig. 8). Taken together, these results reveal a mode of ligand recognition by the Sec14 domain of Kalirin via its surface groove.

## Kal$^{bSec14}$–phospholipid interactions

To determine whether a similar binding mode was employed under more physiologically relevant conditions, we used an in vitro liposome flotation binding assay to evaluate the capability of Kal$^{bSec14}$/Kal$^{Sec14}$ to interact with lipid bilayers of varying composition. A similar assay was used to study the binding of phosphoinositides to the CRAL_TRIO domain of Kalirin preceded by the Ex1C front peptide (Kal$^{cSec14}$)[20]. Unilamellar liposomes formed by mixing synthetic dioleoyl phospholipids and cholesterol in proportions mimicking the membranes of the *trans*-Golgi network (Golgi-mixture) were incubated with Kal$^{bSec14}$. An Accudenz gradient was used to separate liposome-bound proteins, which were recovered from the top of the gradient (fraction 1), from unbound proteins, which remained at the bottom (fraction 4). Protein levels in each fraction were analyzed by Western blotting with an antibody to the Sec14 domain of Kalirin. As observed previously, <2% of the total Kal$^{bSec14}$ bound to the control liposome (Fig. 5a; Lipo).

Lipids to be tested for binding to Kal$^{bSec14}$ were added to the Golgi mixture, accounting for 8% of the total lipid content of the liposomes; although detergents like FC14 cannot be incorporated into liposomes, lysolipids with a single fatty acyl chain attached to the glycerol moiety can be. Taking into account the dimensions and properties of the lipid binding surface groove detected using FC12 and FC14, we first tested two lysophosphatidylcholine (LPC) lipids, 1-lauroyl-2-hydroxy-sn-glycero-3-phosphocholine (LPC12:0) and 1-myristoyl-2-hydroxy-sn-glycero-3- phosphocholine (LPC14:0) (see Supplementary Table 2 for their structures). Both LPC12:0 and LPC14:0 exhibit significant binding to Kal$^{bSec14}$ (Fig. 5). In contrast, LPCs with longer acyl chains (1-palmitoyl-2-hydroxy-sn-glycero-3-phosphocholine, LPC16:0; 1-stearoyl-2-hydroxy-sn-glycero-3-phosphocholine, LPC18:0; 1-oleoyl-2-hydroxy-sn-glycero-3-phosphocholine, LPC18:1) fail to interact with Kal$^{bSec14}$

(Fig. 5). Strikingly, the presence of a second 14-carbon acyl chain (1,2-dimyristoyl-sn-glycero-3-phosphocholine, DMPC) also precludes binding (Fig. 5). Evidently, the single aliphatic hydrocarbon moiety plays an important role in the Kal$^{bSec14}$–lipid interaction.

We next evaluated the ability of liposomes containing lyso-phospholipids with different head groups to interact with Kal$^{bSec14}$. Like LPC, lysophosphoethanolamine (LPE) possesses a positively charged head group (Supplementary Table 2). Liposomes containing 1-myristoyl-2-hydroxy-sn-glycero-3-phosphoethanolamine (LPE14:0) bind Kal$^{bSec14}$ to a level similar to that of LPC14:0 (Fig. 5). On the other hand, liposomes containing 1-myristoyl-2-hydroxy-sn-glycero-3-phospho-(1'-rac-glycerol) (LPG14:0), 1-tridecanoyl-2-hydroxy-sn-glycero-3-phospho-L-serine (LPS13:0), or 1-tridecanoyl-2-hydroxy-sn-glycero-3-phospho-(1'-myo-inositol) (LPI13:0) fail to interact with Kal$^{bSec14}$ (Fig. 5), indicating that the positively charged head groups of LPC and LPE are essential determinants for Kal$^{bSec14}$ recognition. Our previous studies showed that Kal$^{cSec14}$ specifically interacts with liposomes containing L-α-phosphatidylinositol-4-phosphate (PI4P), a phospholipid with two acyl chains and a PI head group[20]. Strikingly, Kal$^{bSec14}$ does not interact with liposomes that contain PI4P (Fig. 5). Similar results were obtained when liposomes containing this same set of test lipids were incubated with Kal$^{Sec14}$ (Supplementary Fig. 9). Together, these data suggest that phospholipids with a single, relatively short acyl chain and a positively charged head group are recognized by Kal$^{bSec14}$ and Kal$^{Sec14}$, and that its front peptide affects its lipid binding specificity.

## LPC lipid binding to the surface groove

We again turned to NMR spectroscopy to determine whether the interaction of Kal$^{bSec14}$ with LPC lipids involves its surface groove; LPE14:0 was precluded from this experiment due to its limited solubility in aqueous solution. As observed using phosphocholine (FC14 and FC12), $^{1}$H and $^{15}$N chemical shifts in Kal$^{bSec14}$ change continuously upon titration with either LPC14:0 or LPC12:0, but not with LPC16:0 (Fig. 6a and Supplementary Fig. 10). The $K_D$ values calculated for the titrations with LPC14:0 and LPC12:0 are $18.9 \pm 2.9$ μM and $316 \pm 26$ μM, respectively (Fig. 6b and Supplementary Fig. 10d); the ~15-fold effect of acyl chain length on binding affinity is similar to that of hydrocarbon chain length with FC14 and FC12. Chemical shift mapping again revealed that the Kal$^{bSec14}$ residues most sensitive to the addition of LPC14:0 cluster around the surface groove (Fig. 6c, d). Twelve residues have significant CSP values (>2σ) while 11 show noteworthy changes, with CSP values >1σ. Interestingly, the side chains of several "2σ" residues (Arg65, Asp69, Arg70, Arg72, Lys106, Asp108, and Lys111) create an ionic surface around the groove opening, while seven "1σ" residues (Val45, Ile71, Leu79, Ile100, Gly104, Leu109 and Leu117) together with three "2σ" residues (Met102, Leu113 and Leu114) form the hydrophobic floor of the groove (Fig. 6e, f).

We next examined the interaction of Kal$^{Sec14}$ with LPC lipids; its affinities for LPC14:0 and LPC12:0 are weaker than those observed for Kal$^{bSec14}$ ($K_D = 57 \pm 8$ μM and $562 \pm 55$ μM, respectively; Supplementary Fig. 11). However, the set of residues in Kal$^{Sec14}$ that exhibit the largest CSPs upon complex formation with LPC14:0 are similar to those identified in Kal$^{bSec14}$ (Supplementary Fig. 12). These results are consistent with the suggested binding mode of FC14/FC12 to Kal$^{bSec14}$/Kal$^{Sec14}$, confirming that the surface groove observed in Kal$^{bSec14}$ serves as a unique ligand binding site not seen in stand-alone Sec14 proteins.

To further probe the importance of its surface groove in lipid binding, we generated two paired Kal$^{bSec14}$ mutants (D69A/R70A and S105A/K106A) that carry alanine substitutions lining each side of the groove. These four amino acid residues were chosen not only because of their large CSPs during NMR titrations, but also for their locations in loops, thus minimizing potential disturbance of protein folding (Fig. 6). Size exclusion chromatography and NMR spectra confirmed that both mutant proteins remained well folded in solution. NMR

titrations of the mutants with LPC14:0 show that Kal$^{bSec14}$D69A/R70A and Kal$^{bSec14}$S105A/K106A exhibit substantially reduced binding affinities to LPC14:0, with $K_D$s of $129 \pm 7$ μM and $244 \pm 21$ μM, respectively (Fig. 7 and Supplementary Fig. 13), as compared to the $K_D$ of $18.9 \pm 2.9$ μM for the wild-type protein. These results strongly support for our conclusion that the surface groove of Kal$^{bSec14}$ serves as the ligand binding site.

As multiple attempts to crystallize Kal$^{bSec14}$ complexed with a LPC lipid failed, we used molecular docking to examine the probability of accommodating LPC14:0 in the surface cavity. LPC14:0 is docked as a flexible entity using AutoDock Vina[28]. Most of its top-ranked docked conformations are projected to bind to Kal$^{bSec14}$ at the surface cavity, although the exact locations and conformations of the docked LPC14:0 molecules differ (Supplementary Fig. 14). Taken together, our structural observations, lipid binding data and mutagenesis studies indicate that Kal$^{bSec14}$ defines a distinct class of Sec14-containing proteins that employ a surface groove to interact with ligands that have a single acyl chain.

## Role of the Ex1B front peptide in LPC binding

Using the structural and NMR titration data for Kal$^{bSec14}$ and Kal$^{Sec14}$, we sought to understand how the Ex1B front peptide might affect lipid binding. The 24 amino-acid Ex1B front peptide, including two proline residues, is negatively charged (Supplementary Fig. 15a). We identified 22 unique peaks in the HSQC spectrum of Kal$^{bSec14}$ but not in that of Kal$^{Sec14}$ (Supplementary Fig. 5c). None of these 22 unassigned peaks were shifted upon titration of increasing amounts of LPC14:0 (Supplementary Fig. 10a), indicating that the Ex1B front peptide does not interact directly with the lipid.

Chemical shift mapping of the superimposed HSQC spectra of Kal$^{bSec14}$ and Kal$^{Sec14}$ revealed that the residues displaying significant CSPs (>2σ) are located in four regions surrounding the rear portion of the surface groove (Supplementary Fig. 15b, c). A network of basic residues located adjacent to these CSP sensitive regions, including Arg52, Lys54, Arg55, Lys93, Arg94 and His126, forms a large positively charged surface patch that could be readily accessed by the Ex1B front peptide (Supplementary Fig. 15d, e). It is thus possible that the Ex1B front peptide affects lipid binding by transiently interacting with the core domain of the protein to change protein dynamics to favor a conformation optimal for ligand binding. In contrast, the positively charged, hydrophobic Ex1C front peptide (Supplementary Fig. 15a) forms an amphipathic helix that interacts directly with phosphoinositides, altering the function of cKal7[20]. Different front peptides may employ distinctive strategies to affect ligand binding. Unfortunately, the limited solubility of the Ex1C front peptide precluded our many attempts to crystallize Kal$^{cSec14}$ or to measure its binding affinity for lipids.

## Structural and sequence comparison to other mammalian SecSR proteins

In addition to Kalirin, the SecSR sub-class of the Sec14 superfamily contains four other mammalian RhoGEFs (TRIO, MCF2/DBL, MCF2L/DBS and MCF2L2) and SESTD1. With the Kal$^{bSec14}$ structure in hand, we searched for sequence and structural features conserved amongst these proteins. First, Kal$^{bSec14}$ shares a much higher degree of sequence identity with SecSR members (~35–77%) than it does with other Sec14 family proteins (Supplementary Fig. 16a). The residues engaged in specific phosphatidylinositol head group binding in Sec14p and its close homologs[12] are not conserved in SecSR family members. Second, as for Kalirin, each SecSR protein lacks a predicted CRAL_TRIO_N domain. Third, like Kalirin, isoforms of TRIO, MCF2L and MCF2L2 include an N-terminal front peptide preceding their CRAL_TRIO domains (Supplementary Fig. 16b). As observed for *KALRN*, alternative splicing generates isoforms of TRIO and MCF2L with front peptides that have very different biophysical properties. For example, in

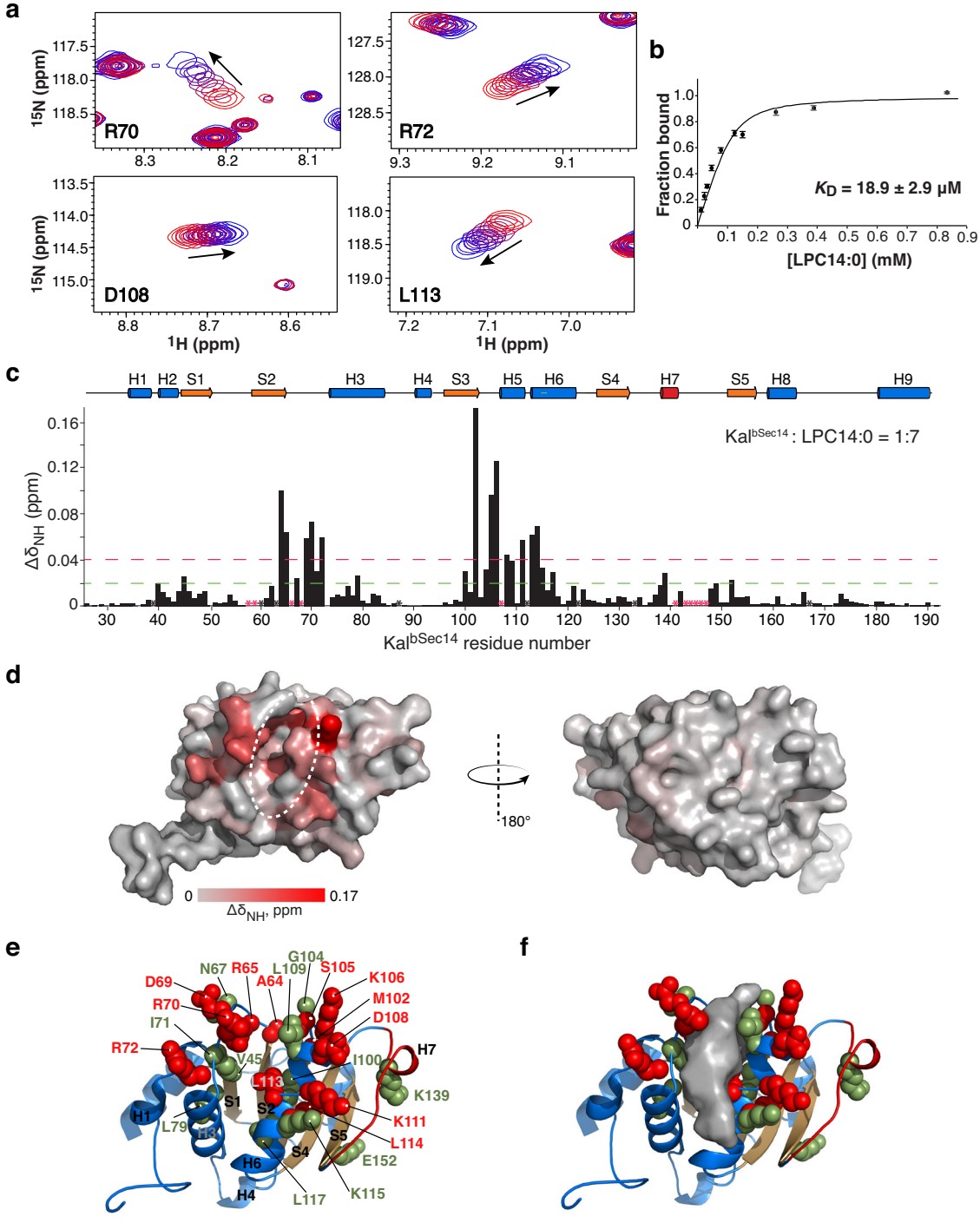

**Fig. 6 | LPC14:0 binds to the surface groove of Kal^bSec14. a** Four residues of Kal^bSec14 that demonstrate CSPs when titrated with increasing amounts of LPC14:0 (1:7 final molar ratio). For each residue, the cross peaks are color-ramped from red to blue with increasing LPC14:0 concentrations, as indicated by arrow. The full series of spectra for the titration are shown in Supplementary Fig. 10a. **b** A plot of normalized global fitting of the averaged CSPs ($\triangle\delta_{obs}/\triangle\delta_{max}$) as a function of LPC14:0 concentration to estimate the $K_D$ for binding. The fitting data and the $K_D$ value represent the mean ± S.D. of the CSP data for individual residues ($n = 13$). Source data are provided as a Source Data file. **c** Plot of per-residue backbone CSPs between free and LPC14:0-bound states of Kal^bSec14. Proline residues and residues missing backbone assignment are indicated by asterisks (black, proline; red,

unassigned). Dashed green and red lines indicate CSP values within one (1σ) and two (2σ) S.D. of the average CSP (0.02 ppm) among all assigned residues, respectively. Source data are provided as a Source Data file. **d** Surface representation of Kal^bSec14 colored according to CSPs induced by LPC14:0 binding, from light gray (no observed CSP) to red (maximum CSP). Dashed ellipse indicates the surface groove. **e** Residues with large LPC14:0-induced CSPs mapped on the structure of Kal^bSec14; the structure is shown in an orientation similar to that in the left side of **d**. Residues with CSPs >1σ but <2σ and >2σ are shown as green and red spheres, respectively. **f** Surface representation of the surface groove surrounded with the residues specified in **e**.

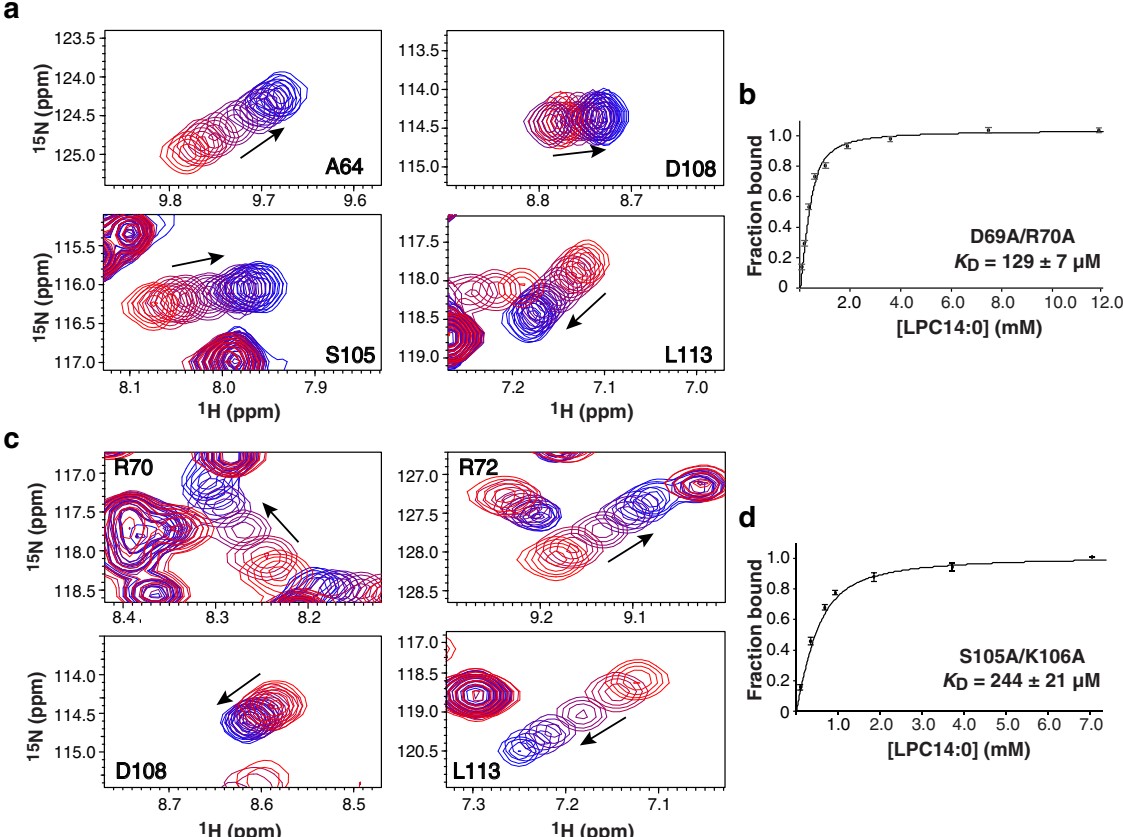

**Fig. 7 | Effects of the surface groove mutations of Kal[bSec14] on LPC14:0 binding.** **a, c** Titration data were obtained for two Kal[bSec14] mutant proteins, each bearing mutations at two positions, D69A/R70A (**a**) and S105A/K106A (**c**). For each mutant, data are shown for four residues that demonstrate CSPs when titrated with increasing amounts of LPC14:0 (1:24 and 1:15 final molar ratio for Kal[bSec14]D69A/R70A and Kal[bSec14]S105A/K106A, respectively); cross peaks are color-ramped from red to blue with increasing LPC14:0 concentrations, as indicated by arrow. The full series of spectra for the titration are shown in Supplementary Fig. 13a, b. **b, d** Plots of normalized global fitting of the averaged CSPs ($\triangle\delta_{obs}/\triangle\delta_{max}$) as a function of LPC14:0 concentration to estimate the $K_D$ for binding Kal[bSec14]D69A/R70A (**b**) and Kal[bSec14]S105A/K106A (**d**). The fitting data and the $K_D$ values represent the mean ± S.D. of the CSP data for individual residues for Kal[bSec14]D69A/R70A ($n = 9$) and Kal[bSec14]S105A/K106A ($n = 18$). Source data are provided as a Source Data file.

isoforms 4 and 9 of MCF2L, the CRAL_TRIO domain is preceded by a peptide predicted to form an amphipathic helix[29]. Fourth, secondary-structure and structure predictions suggest that the overall fold of the CRAL_TRIO domain and the first helix of the neighboring SR of these proteins resembles the fold observed for Kal[bSec14] (Supplementary Fig. 16c). The structures predicted using AlphaFold[30] share a 1.3–1.7 Å rmsd with Kal[bSec14] for equivalent $C_\alpha$ positions (Supplementary Fig. 16a). Last, residues exhibiting large ligand-induced CSPs in Kal[bSec14] are generally conserved, maintaining their hydrophobicity, charge or size in the other SecSR RhoGEFs and in SESTD1, suggesting formation of a similar ligand-binding site (Supplementary Fig. 16b). Collectively, our data identify Kal[bSec14] as an example for a distinct class of Sec14-like domains with ligand binding sites compatible with their roles in dynamic signaling pathways.

## Discussion

Structural studies of multiple stand-alone Sec14-domain proteins have been integral in understanding how ligand binding induces specific biological outcomes within this large and diverse protein family. Sec14 domains also play essential roles in a number of highly-conserved multidomain mammalian RhoGEFs known to participate in the protein complexes that control signaling, membrane trafficking and cytoskeletal organization. The results presented here provide structural and biochemical evidence that Kalirin possesses a CRAL_TRIO domain with a unique ligand binding site. Preceded by alternate front peptides and followed by multiple SRs, the Sec14 domain of Kalirin contains structural and functional features evolving from both CRAL_TRIO and BCH sub-group proteins and serves as a prototype for other SecSR sub-class family members.

Crystallographic analysis reveals that Kal[bSec14] exhibits clear structural homology to the αβα fold of canonical CRAL_TRIO domains; a similar fold is observed in the BCH domain of p50RhoGAP even though one of its five strands is contributed by the other monomer present in its intertwined dimer[15]. However, Kal[bSec14] adopts a closed conformation in which H7, the presumed gate helix, blocks access to its fragmented hydrophobic interior, leading to the discovery of its alternate ligand binding site. By assigning NMR resonances of Kal[Sec14], we developed a powerful tool to study the interactions of Kal[bSec14]/Kal[Sec14] with ligands in solution. Our NMR CSP data and mutagenesis studies demonstrate that the residues lining a distinct surface groove of the Kalirin Sec14 domain are responsible for its interaction with selected detergents and LPC lipids. Pocket distribution analyses using the structures of six CRAL_TRIO sub-family proteins confirm the notion that each contains an internal hydrophobic pocket and lacks an exposed groove similar to that observed in Kal[bSec14]. Two exposed pockets were identified in the BCH domain of p50RhoGAP, one in a location similar to that of Kal[bSec14] and the other close to the canonical gating helix (PDB ID: 7E0W; Supplementary Fig. 17) A tetraethylene glycol (TEG) molecule co-purified with the protein occupies the first pocket, in a straight up pose with one end diving into the hydrophobic core and the other projecting out of the surface. Our comparison of the CRAL_TRIO domains of Kal[bSec14], BCH and other Sec14 proteins

suggests that the ligand binding pocket in the BCH domain of p50RhoGAP is an intermediate form between the exposed surface cavity in Kal[bSec14] and the buried pocket in canonical CRAL_TRIO domains.

Unlike many stand-alone Sec14 domain proteins, the Kalirin CRAL_TRIO domain lacks a helical CRAL_TRIO_N lobe. In both phosphatidylinositol- and phosphatidylcholine-bound structures of yeast Sfh1, the lipid head groups are oriented towards the N-terminal lobe while the acyl chains remain sequestered in the hydrophobic pocket[11]. The residues engaged in lipid head group binding (i.e. binding barcodes) are located in both the CRAL_TRIO_N and CRAL_TRIO domains and are conserved in Sec14p homologs[12] and in α-tocopherol transfer protein[25], indicating that these two domains function as a unit. As both the CRAL_TRIO_N domain and the phospholipid binding barcodes are absent from Kalirin, our structural and biochemical studies are consistent with the inability of Kal[bSec14] to recognize phosphatidylinositides[20]. Kal[bSec14] binds LPC12:0 and LPC14:0 about twice as tightly as Kal[Sec14]. Comparisons of the NMR spectra of Kal[bSec14] and Kal[Sec14] indicate that the Ex1B front peptide might mimic at least some of the functions attributed to CRAL_TRIO_N domains by populating the Kal[bSec14] core fold in a conformation optimal for ligand binding.

Our search for lipids that bind to Kal[bSec14] and Kal[Sec14] led to the identification of lysophospholipids with a single, relatively short acyl chain and a positively charged head group, with $K_D$s in the micromolar ranges. The ~20 μM $K_D$ of Kal[bSec14] for LPC14:0 is distinctly different from the nanomolar $K_D$ values determined for the interactions of stand-alone Sec14 proteins such as α-TTP/Sec14p/SPF/CRALBP and their ligands[31]. Identification of an exposed groove suggests a major role for protein-lipid interactions on the Kal[bSec14] surface, rather than lipid transfer activity via the canonical buried hydrophobic pocket observed in stand-alone Sec14 proteins[11].

It is widely recognized that rapid responses require low affinity (1–100 μM), readily reversible interactions amongst multiple proteins; modularity of recognition domains and the simultaneous use of multiple low-affinity interactions ensures more readily modified, specific interaction[32–36]. Functional studies using knockout mice and knockdown technologies indicate that Kalirin can play a role in downstream signaling from receptor tyrosine kinases, GPCRs and ion channels[17–20]. Consistent with these observations, tissue-specific, regulated phosphorylation of multiple Ser/Thr and Tyr residues in Kalirin has been documented[17–20]. In addition, multiple protein-protein interactions involving the Sec14, SR, PH and SH3 domains of Kalirin have been identified, along with interactions of its short, linear peptide motifs with PDZ-, SH3- and SH2-domains. Coupled with these additional interactions, the low-affinity interaction of Kal[bSec14] with lysolipids may contribute to the essential role the Kalirin Sec14 domain plays in controlling spine length and endocytic trafficking[18–20]. While the SRs and PDZ-binding motif of Kal7 may ensure its stable enrichment in the post-synaptic density, changes in the lipids accessible to the Kalirin Sec14 domain at the post-synaptic membrane or in tension on its SRs may contribute to the inhibitory effect these regions have on its GEF activity (see below).

For the vast majority of Sec14 superfamily members, the identities of their physiologically relevant ligands remain unknown[3]. Lysophospholipids, integral components of the plasma membrane, are usually present only in small amounts[37]. They can be generated rapidly by phospholipases (A1 and A2), serving as intermediates in phospholipid metabolism and as second messengers in GPCR-mediated signaling pathways[37–39]. With a relatively large hydrophilic head group and a cone shape, incorporation of lysolipids into membrane bilayers introduces packing stresses and drives lipid assemblies into structures with more positive curvature[40,41]. Interestingly, it should be noted that Kal[bSec14] and Kal[Sec14] bind selectively to a very narrow range of the LPC species (with hydrocarbon tails from 12:0 to 14:0). While our data

suggest a role for phospholipase activation in generating a lysophospholipid that might bind to Kal[bSec14], acyl chains with 14 carbons are generally not prevalent. Whether membrane curvature or membrane disruption associated with vesicular trafficking might expose a limited region of longer acyl chains has not yet been assessed. New methods are needed to allow the assessment of lipid/protein interactions in a cellular environment with subcellular specificity[32,42].

Our structure-based sequence alignment analyses demonstrate that the Kalirin CRAL_TRIO domain can serve as the prototype for a distinct functional subclass of the Sec14 superfamily. The six members of the SecSR sub-group are multidomain proteins with an N-terminal CRAL_TRIO domain followed by a variable number of SRs. Kalirin, TRIO, MCF2, MCF2L and MCF2L2 are Dbl family RhoGEFs, while SESTD1 partners with multiple GAPs and GEFs to modulate RhoA GTPase activities[43]. Unlike conventional CRAL_TRIO domain containing proteins, SecSR family members share high sequence homology and are implicated in control of actin cytoskeletal organization and dynamics by modulating the activity of small GTPases. None of the other Sec14 sub-groups contains a RhoGEF, and RhoGAPs are found exclusively in BCH domain sub-families[1]. Besides a role in subcellular localization, the function of the SecSR domain in five RhoGEF proteins seems to be inhibition of GEF activity. The catalytic activity of the isolated GEF domain of Kal7 is much higher than that of intact Kal7 or Kal7 lacking its Sec14 domain and first four SRs[17]. In agreement with this observation, Sec14 domain and SR removal reveals the full transforming activity of MCF2 and MCF2L, identifying an autoinhibitory intramolecular interaction of the Sec14 domain with the GEF PH domain[44,45]. A subset of BCH-domain containing RhoGAPs, much like their GEF counterparts, use an autoinhibitory interaction of their BCH and GAP domains to control activity[15,46,47]. As the postulated non-protein ligands for the SecSR-containing proteins are currently unknown, an exciting possibility is that the SecSR domain may directly interact with the small GTPase or its post-translationally modified protein terminus to regulate signaling cascades. Further structural and functional studies are needed to understand the potential implications for small GTPase signaling mediated by the previously uncharacterized SecSR domain-containing proteins.

## Methods

### Protein expression and purification
The Kalirin constructs were designed based on the rat *KALRN* sequence [NCBI Accession #: U88156.1 with two single nucleotide sequence corrections (C197G and A244T) that lead to two amino-acid changes (T66S and N82Y)]; constructs were verified by DNA sequence analysis. Amino acid numbers are based on the initiator Met encoded by Ex1B as residue 1. Inserts encoding Kal[bSec14] (residues 2–192; 21.7 kDa) and Kal[Sec14] (residues 25–192; 19.4 kDa) were cloned into a modified pET15b vector containing a removable tobacco etch virus (TEV) protease recognition site. Primer sequences are: 5'-ATATGCGGCCGCAACCCCC CGGAGGGGGCA (Kal[bSec14]-Fwd), 5'-ATATGGTACCTCACAGGGAGAGG CGCAACTCAA (Kal[bSec14]-Rev), and 5'-ATATGCGGCCGCGGATCTT TTCGGAATGATGGTTT (Kal[Sec14]-Fwd). The surface groove mutations of Kal[bSec14] were introduced using an overlap PCR method and confirmed by sequencing. Each of the constructs was expressed in *E. coli* BL21(DE3) and purified by Ni[2+]-nitriloacetic acid (GE Healthcare) affinity chromatography. After removal of the N-terminal His₆ tag by overnight cleavage with TEV protease, the proteins were further purified by anion-exchange (Source Q; GE Healthcare) and gel-filtration (SD200; GE Healthcare) chromatography. For crystallization, Kal[bSec14] was concentrated to 1.1 mM by ultrafiltration in a buffer containing 10 mM Tris-HCl (pH 7.6), 150 mM NaCl, and 5 mM dithiothreitol (DTT). SeMet-substituted Kal[bSec14] was produced following established procedures[48] and purified as described above. For NMR chemical shift perturbation experiments, each protein was expressed in M9 medium supplemented with $^{15}NH_4Cl$ (1 g/L; Cambridge Isotope Laboratories) as

the sole nitrogen source. For NMR backbone resonance assignment experiments, Kal$^{Sec14}$ was expressed in M9 medium prepared with $^{15}NH_4Cl$ and [$^{13}C$]-D-glucose (2 g/L; Cambridge Isotope Laboratories) as the sole nitrogen and carbon sources, respectively. The $^{15}N$- and $^{15}N$/$^{13}C$-labeled proteins were purified as described above and concentrated to 0.4 mM in NMR buffer containing 20 mM Na/K phosphate (pH 7.2), 120 mM NaCl, 2 mM β-mercaptoethanol, 0.2 mM EDTA, 8% $D_2O$ and 0.02% $NaN_3$, immediately prior to data acquisition.

## Crystallization and structure determination

Kal$^{bSec14}$ was crystallized in a solution consisting of 100 mM sodium cacodylate (pH 6.5), 0.2 M ammonium sulfate, 20%-25% PEG 3350, 0.2 M NaCl at 4 °C using the hanging-drop vapor diffusion method. Crystals were cryoprotected in reservoir solution supplemented with glycerol to a final concentration of 10% and flash-cooled in liquid nitrogen. Diffraction data were collected using the Life Science Data Collection (LSDC) software at the Stanford Synchrotron Radiation Lightsource (SSRL) beamlines 14-1 and 12-2 and 17-ID-1 and 17-ID-2 of National Synchrotron Light Source II (NSLS-II). The diffraction data were processed with autoPROC[49] and Fast DP[50,51]. The crystals contain eight molecules in the asymmetric unit. The structure of Kal$^{bSec14}$ was determined by single-wavelength anomalous dispersion using the data collected at the selenium peak wavelength. About 16 selenium sites were located by HySS as implemented in AutoSol/PHENIX[52]. Initial phases calculated from these sites were improved by density modification using Phaser/PHENIX[52]. The resulting electron density map was readily interpretable and used to build >90% of the molecules with the program COOT[53]. Iterative cycles of refinement using REFMAC5/CCP4[54] and BUSTER (Global Phasing Limited) followed by manual rebuilding in COOT were carried out until no further improvement of the R$_{free}$ factor was observed. X-ray data collection and phasing and refinement statistics are summarized in Supplementary Table 1. Ramachandran statistics were calculated using MolProbity[55]. Molecular graphics were rendered using PyMOL (Schrödinger LLC).

## NMR resonance assignment

NMR spectra for backbone assignment were recorded using VnmrJ (https://openvnmrj.org/) at 25 °C on an Agilent VNMRS 800 MHz ($^1H$) spectrometer equipped with a cryogenic probe. A set of standard double and triple resonance experiments were performed using 0.4 mM uniformly $^{13}C$/$^{15}N$-labeled Kal$^{Sec14}$ in NMR buffer. The spectra used for sequence specific assignment ($^1H$, $^{15}N$, $^{13}C_\alpha$, $^{13}C_\beta$, and $^{13}C_O$) included 2D $^1H$-$^{15}N$ HSQC, and 3D HNCO, HNCACB, HN(CO)CACB, HN(CA)CO and $^{15}N$-filtered NOESY-HSQC[56–58]. All spectra were processed with NMRPipe[59] and analyzed using CARA (http://cara.nmr.ch) and CcpNmr Analysis[60], which were made available through NMRbox[61]. The near-complete resonance assignment was obtained with 92.7% of all expected chemical shifts assigned (93% HN, 88% N, 91% C$_O$, 92% C$_\alpha$, 88% C$_\beta$). The assignments for Kal$^{Sec14}$ peaks in the $^1H$-$^{15}N$ HSQC spectrum were readily transferred to the cross peaks for Kal$^{bSec14}$. Secondary structure predictions for Kal$^{Sec14}$ were performed using chemical shift assignments of five atoms (HN, N, C$_O$, C$_\alpha$ and C$_\beta$) for a given residue in the sequence with TALOS-N[62]. The predicted ρ(α) and ρ(β) values for each residue are provided in the Source Data file.

## NMR titration experiments

$^1H$-$^{15}N$ HSQC spectra of the uniformly $^{15}N$-labeled Kal$^{bSec14}$ and Kal$^{Sec14}$ were recorded at 25 °C on an Agilent VNMRS 800 MHz ($^1H$) spectrometer equipped with a cryogenic probe. $^1H$-$^{15}N$ HSQC spectra of the uniformly $^{15}N$-labeled Kal$^{bSec14}$ mutants were recorded using the Top-Spin software (https://www.bruker.com/en/products-and-solutions/mr/nmr-software/topspin.html) at 25 °C on a Bruker Avance NEO 600 MHz spectrometer equipped with a cryogenic probe. All detergents and lipids used in this study were purchased from Cube Biotech Inc. (Wayne, Pennsylvania) and Avanti Polar Lipids, Inc. (Alabaster,

Alabama), respectively; details are listed in Supplementary Table 2. After collecting an initial protein-only spectrum with a starting protein concentration of 0.4 mM, each detergent or lipid was dissolved in the same NMR buffer and gradually added to the protein until no further CSP was observed. Each series usually comprised 8-12 titration points, with ligand concentrations ranging from 0 to 8 mM and final protein:ligand ratios from 1:7 to 1:25 (the protein concentration in each mixture during titration was confirmed by UV-vis absorbance at 280 nm). The HSQC spectrum was recorded for each titration point using the same instrument setup as that of the apo protein. All spectra were processed with NMRPipe[59], and peak heights and locations were analyzed in CcpNmr Analysis[60].

For each titration series, observed CSPs for a single amino acid residue were calculated using the combined shift change (Δδ) of the backbone amide nitrogen and proton based on the equation $\triangle\delta\,(ppm) = \sqrt{[\triangle\delta_H^2 + (\alpha \cdot \triangle\delta_N)^2]}$, where $\Delta\delta_H$ and $\Delta\delta_N$ are the $^1H$ and $^{15}N$ chemical shift differences between the free and bound states, respectively, and $\alpha$ denotes the chemical shift scaling factor for $^{15}N$[63]. The scaling factor $\alpha$, which was set to 0.154[64] for this study, was determined from the ratio of the average variances of the $^1H$ and $^{15}N$ chemical shifts observed for the 20 common amino acid residues in proteins as deposited with the Biological Magnetic Resonance Bank (BMRB)[65].

In each titration series, the peaks for all residues were ranked based on the CSP observed upon titration at the highest ligand concentration. For those peaks that showed a significant CSP, defined as those larger than two standard deviations ($2\sigma$) of the average shift for all residues, nearest neighbor chemical shift difference analysis using custom python scripts was performed to confirm shifted peak locations. The well-resolved and non-overlapped CSPs, usually including most of the $2\sigma$ CSPs and some of the $1\sigma$ CSPs, were then plotted as a function of the ligand concentration and the dissociation constant ($K_D$) was extracted by nonlinear least-square fitting of the following equation: $\triangle\delta_{obs} = \triangle\delta_{max} \frac{(K_D + [L]_t + [P]_t) - \sqrt{(K_D + [L]_t + [P]_t)^2 - 4[L]_t[P]_t}}{2[P]_t}$, where $[P]_t$ and $[L]_t$ are the total protein and ligand concentrations, respectively, and $\triangle\delta_{max}$ is the chemical shift difference at saturation[66]. The bound fraction plotted as the y-axis in the Figures was defined as $\triangle\delta_{obs}/\triangle\delta_{max}$ based on the average of the normalized CSPs from all the selected peaks. Titration data for each residue and each peak used for $K_D$ calculations are provided in the Source Data file.

## Molecular docking

Three-dimensional coordinates of the lipids were generated by Open Babel[67] using the ChemDraw files downloaded from the website of Avanti Polar Lipids (Alabaster, Alabama). Docking of the lipids into the Kal$^{bSec14}$ structure was performed using the iterated Local Search Global Optimization algorithm provided by AutoDock Vina[68]. The PDBQT format files (required as input) of both the lipids and Kal$^{bSec14}$ were generated using the AutoDock Tools package provided by AutoDock 4. The flexible ligand docking protocol was adopted in this work to allow rotation of any rotatable bonds in the lipid when it is docked to the binding site in the protein. Once the rotatable bonds are identified, the ligand is allowed to be flexible by moving different rigid parts through the iterated local search global optimizer. During the global energy optimization, the ligand binding score is evaluated by an empirical scoring function, which is a weighted sum of terms representing van der Waals, hydrogen bonds, electrostatic and desolvation contributions. The entire surface of the Kal$^{bSec14}$ structure was searched for possible binding sites without bias. All other parameters were set at the default values defined by AutoDock Vina. In particular, a cutoff distance of 5 Å for van der Waals interactions and 3.7 Å for hydrogen bonds between ligand atoms and protein acceptors were used. The resulting poses were scored based on the free energy of binding in which the least energetic

poses would be ranked as the top hits. Total ten hits were outputted and top four hits were shown in Supplementary Fig. 14.

## Liposome preparation and flotation assays

The synthetic and natural lipids used in flotation assays were purchased from Avanti Polar Lipids, Inc. (Supplementary Table 2). All powdered lipids were dissolved in chloroform per the manufacturer's instructions. Liposomes were prepared as previously described[20]. In brief, stocks of synthetic 1,2-dioleoyl-sn-glycero-3-phosphocholine (DOPC), -phosphoethanolamine (DOPE), and -phospho-L-serine (DOPS) were mixed with cholesterol in molar proportions mimicking those observed in secretory pathway membranes (control lipid mix). Individual test lipids were dried together with the control lipid mix under a nitrogen stream and lyophilized overnight. Each dried lipid film was reconstituted to a final concentration of 1 mM in HN buffer containing 20 mM HEPES (pH 7.4) and 150 mM NaCl. Each final lipid mixtures consisted of 52% DOPC, 20% DOPE, 5% DOPS, 15% cholesterol and 8% tested lipid; inclusion of a trace amount of fluorescent lipid (Liss Rhod PE) allowed for localization of liposomes following gradient fractionation (see below). Each lipid mixture underwent five freeze-thaw cycles separated by vigorous vortexing each time it was thawed. Using a mini-extruder, unilamellar liposomes were prepared by passing each lipid mixture 21 times through a polycarbonate membrane of 100-nm pore size. Hydrated lipid mixtures were stored at −20 °C for up to 1 month; liposomes were freshly prepared for each experiment.

Flotation assays were conducted following the protocol previously described[20]. Accudenz (Accurate Chemical and Scientific Corp., Westbury, NY) solutions (80 and 20% weight/vol) were prepared in HN buffer. To remove any particulate material, the protein to be tested was first centrifuged at 436,000 g at 4 °C for 15 min in a TL-100 ultracentrifuge using a Beckman TLA100.2 fixed angle rotor. Freshly prepared unilamellar liposomes (50 µl) were combined with 0.2 µg of the centrifuged protein and incubated at room temperature for 30 min. The entire liposome/protein mixture was then combined with 50 µl of 80% Accudenz at the bottom of an ultracentrifuge tube and carefully mixed to completion by pipetting. The resulting mixture was overlayered with 250 µl of 20% Accudenz and then with 50 µl of HN buffer. Samples were centrifuged at 259,000 g at 4 °C for 30 min in the same ultracentrifuge using a Beckman TLS55 swinging bucket rotor. Gradients were then divided into four 100-µl fractions, which were carefully collected from the bottom of the tube using gel loading micropipette tips. Each fraction was analyzed for liposome and protein content using spectrophotometry (546-nm excitation/571-nm emission) and Western blot, respectively. Only those assays with at least 90% of the Liss Rhod PE marker recovered in the top fraction (fraction 1) were used for Western bloting analysis.

For Western bloting analysis, equal aliquots of each gradient fraction were denatured in 1× Laemmli sample buffer, fractionated on 15% acrylamide gels, and transferred to PVDF membranes. The protein content was detected using a custom-made rabbit antibody to the Sec14 domain of Kalirin (CT302[69], 1:3000) and an HRP-tagged goat anti-rabbit IgG (H + L) secondary antibody (catalog # 31460, 1:6000; ThermoFisher Scientific). SuperSignal West chemiluminescent substrate (Thermo Scientific) and a Syngene PXi digital imaging system (Synoptics Group) were used to visualize blots, with exposures adjusted to be in the linear range. After background subtraction, bands were quantified using the GeneSys software provided with the imaging system, and data were represented as a proportion of total protein recovered. Uncropped blots, quantification of each data point and exact P-values are provided in the Source Data file.

## Reporting summary

Further information on research design is available in the Nature Portfolio Reporting Summary linked to this article.

## Data availability

The data that support this study are available from the corresponding authors upon reasonable request. The atomic coordinates and structure factors of Kal[bSec14] have been deposited in the Protein Data Bank (PDB) under accession code 7UR2 (Sec14 domain of Kalirin). The backbone ¹H, ¹³C and ¹⁵N resonance assignments of Kal[Sec14] have been deposited to BMRB under the accession number 51382 (NMR backbone assignments of the Kalirin Sec14 domain). Atomic coordinates for the previously determined Sec14 domain containing proteins were accessed from Protein Data Bank under accession codes 1AUA, 1OIZ, 1O6U, 2D4Q, 3B7N, 3HY5, 3W67, 6W32, and 7E0W. All other supporting data are available within the article and its Supplementary Figures and Tables. The source data underlying Figs. 3–7, Supplementary Figs. 6–11 and Supplementary Fig. 15 are provided as a Source Data file. Specific data P-values are also included within the Source Data file. Source data are provided with this paper.

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

## Acknowledgements

We thank W. Shi, J. Jakoncic and A. Soares at the beamlines 17-ID-1 and 17-ID-2 of NSLS-II for assistance with the X-ray data collection, I. Bezsonova, D. Korzhnev and M. Maciejewski for NMR data collection and processing, and A. K. Pozhidaeva and P. Setlow for critical reading of the manuscript. This work was supported by National Institutes of Health (NIH) Grants DK032948 (to R.E.M. and B.A.E.), GM123249 (to J.C.H.), GM099948 (to B.H.) and GM135592 (to B.H.). This study made use of NMRbox: National Center for Biomolecular NMR Data Processing and Analysis, a Biomedical Technology Research Resource, which is supported by NIH Grant P41GM111135 (to J.C.H.). Support was also provided by the Dr. Daniel Schwartzberg Fund (to R.E.M. and B.A.E.). This research used resources of NSLS-II, a U.S. Department of Energy (DOE) Office of Science User Facility operated by Brookhaven National Laboratory under Contract No. DE-SC0012704. The Center for BioMolecular Structure (CBMS) is primarily supported by the NIH, National Institute of General Medical Sciences (NIGMS) through a Center Core P30 Grant (P30GM133893), and by the DOE Office of Biological and Environmental Research (KP1605010). Use of the SSRL, SLAC National Accelerator Laboratory, is supported by the DOE Office of Basic Energy Sciences under Contract No. DE-AC02-76SF00515. The SSRL Structural Molecular Biology Program is supported by the DOE Office of Biological and Environmental Research, and by the NIGMS (P30GM133894). The contents of this publication are solely the responsibility of the authors and do not necessarily represent the official views of NIGMS or NIH.

## Author contributions

R.E.M., B.A.E. and B.H. conceived the study. Y.L., T.I.D. and B.H. performed the crystallographic studies., Y.P., Y.L. and B.H. performed the NMR analyses. Y.L. and B.H. performed the biochemical experiments. Y.P. and J.C.H. provided the NMR data analysis resources. All authors analyzed the results. B.A.E. and B.H. wrote the manuscript. All authors edited and approved the final manuscript.

## Competing interests

The authors declare no competing interests.
