## [Peer Review File · Nature Communications]

Structure of the Sec14 domain of Kalirin reveals a new class of lipid-binding module in RhoGEFsReviewers' Comments:

Reviewer #1:

Remarks to the Author:

This work presents novel and valuable insights into the research field of the Kalirin proteins. The 3D structure of the CRAL-TRIO domain in the gene product of the KALRN gene is reported for the first time. Furthermore, the authors have identified that the binding mode of KalbSec14 and KalSec14 to its ligand is not mediated through the classical bindings mode known in CRAL-TRIO proteins. Classically the binding occurs through the canonical binding pocket reported in many other CRAL-TRIO proteins. They analyzed the crystal structure for a supposed internal binding pocket but realized it is missing in KalbSec14. This observation is reinforced by in silico analysis that revealed the presence of small discontinuous inner volumes that are not comparable in size and shape to typically reported binding pockets in CRAL-TRIO domains. Additionally, the spatial orientation of several helices in KalbSec14 and KalSec14 impedes the access to the supposed internal pocket via a lid. It hence adopts a closed holo-conformation of the CRAL-TRIO proteins. In addition, the authors have identified a locus on the surface that forms an elongated groove featuring a dipole-like surface charge that preferentially binds phospholipids with positively charged head groups such as phosphatidylethanolamine and phosphatidylcholines (PE & PC). The authors compared the superimposed crystal structure of KalbSec14 to that of p50RhoGAP (PDB ID: 7E0W), claiming that the groove in KalbSec14 overlaps with the binding locus of tetraethylene glycol (TEG) in the BCH domain of p50RhoGAP. They hint that this observation further proves that the groove is a novel binding site in the CRAL-TRIO domain of KalbSec14. However, after a thorough analysis consisting of a structural and topological (see figure 1) comparison to other Sec14-like proteins, including α -TTP, CRALBP, and Sec14p, it seems that the groove in KalbSec14 is a variation of the canonical binding pocket rather than a novel unique ligand-binding site. The structure of p50RhoGAP (PDB ID: 7E0W) was compared to α -TTP (PDB ID: 3W67), CRALBP (PDB ID: 3HY5), and Sec14p (PDB ID: 1AUA) to verify and visually inspect the position of TEG in a broader scope within the CRAL-TRIO domains. The surface groove does not necessarily represent a new, previously unknown binding site. Please see the attached figure 2 (panels A, B, and C) – TEG's location coincides with the canonical binding pocket in all three proteins. This observation is particularly striking when it is compared to α -TTP, where TEG and α -tocopherol are spatially congruent. The groove is embedded between helix 3 and 6 of KalbSec14, as shown in figure 2C. In the other CRAL-TRIO proteins, these analogous helices (see figures, orange helices) are also part of the canonical binding pocket. However, the groove between these helices is covered by helix 4 (see figures, green helix) of the CRAL-TRIO-N domain in the Sec-14-like protein featuring that N-terminal segment. In the case of KalbSec14, the groove is accessible from the bulk solvent since it is missing the CRAL-TRIO-N part. With all this evidence, the authors must review their structure in a more thorough comparison and re-think their conclusion regarding the binding groove as a novel site. There is strong evidence that the groove is part of the known binding site featuring a novel binding mode.

Furthermore, the authors have systematically and rationally characterized the binding of KalbSec14 and KalSec14 to several lipids. The before-mentioned groove was tested for its binding ability to several PC variations. NMR chemical shift perturbations (CSP) were used to identify potential ligands and the amino acid residues involved in the binding interactions. They determined that artificial PCs (LC12 & LC14) featuring a 12 and 14 carbon long acyl chain showed the highest binding compared to the others. They also identified that the amino acids affected mainly by the CSP upon ligand binding were indeed those lining both sides of the groove recognized in the crystal structure. This evidence further supports their conclusion that the groove represents a binding site for lipids. In a second experiment, under more physiologically relevant conditions, the authors used standardized liposomes mimicking the membrane of the trans-Golgi network supplemented with PC variants of interest to determine whether KalbSec14 binds to it. They separated liposome-bound proteins by an Accudenz gradient and analyzed protein levels in each fraction by Western blot. In agreement with the previous experiment, most bound protein was recovered from liposomes containing LPC 12:0 and LPC 14:0. Intriguingly, little binding was observed when DMPC was used, featuring two 14-carbon acyl chains

instead of one, but no comment on possible reasons for this observation is postulated.

The finding that preferentially lipids with positively charged headgroup and featuring only one short acyl chain (12 and 14 carbons) is well documented in figure 5. This observation hints at a specific binding that recognizes distinct features of the lipids. However, the reported binding affinities are relatively weak, featuring low to middle micromolar range values. Such high dissociation constants do not unanimously prove that LPC 12:0 and LPC 14:0 are physiologically relevant ligands of KalbSec14 and KalSec14. Especially when compared to previously reported K_d of known physiological ligands in the CRAL-TRIO protein family 1. Specific bindings are reported in the two digits nanomolar range (e.g., α -TTP/ α -tocopherol: 25 nM) and non-specific bindings in the one-digit micromolar scope (e.g., α -TTP/Trolox: 1,004 μ M).

Whereas the reported affinities in this paper for LPC 12:0 and LPC 14:0 to KalbSec14 are 18.9 μ M and 316 μ M, respectively. And in the case of KalSec14, it is 57.2 μ M and 562 μ M for LPC 12:0 and LPC 14:0, respectively. These values represent approximately a 20 to 500-fold weaker binding affinity of the best binding LPC to KalbSec14 compared to a synthetic non-physiological molecule as Trolox to α -TTP.

This controversial finding is quite puzzling; either the binding affinities are wrong, and another measurement method should be employed to cross-check the values, or the binding is not that specific, hence wise questioning its physiological relevance. The authors do not discuss the meaning of the reported low affinities nor their implications in a physiological context.

The authors concluded that KalbSec14 binds preferentially lipids featuring a positively charged head group such as PC or PE and a single relatively short 14-carbon acyl chain. They claim this is a new and unique ligand-binding site that employs a surface groove not seen in stand-alone Sec14 proteins. However, I miss the physiological evidence for some conclusions proposed by the authors. Primarily, the meaning of the binding of the proposed ligands (LPC 12:0 and LPC 14:0) is not elucidated enough. The physiological role of these lipids is not discussed. Therefore, I'm surprised by the bold wording in some passages. I don't think that this paper has provided enough insight to make such conclusive statements.

Nevertheless, it is an essential paper to understand the function and mode of action of Karilins. The elucidation of the first crystal structure of the CRAL-TRIO domain in Kalirin is a significant finding that will positively promote this research field. Furthermore, the experiments were carried out carefully and correctly. The data and its analysis are also meticulous. All experiments are sufficiently documented so that other experimenters can reproduce the experiments without further ado. All in all, this paper is extensive and solid work; the only flaw I see is the wording in some conclusions that should be attenuated.

For the reasons mentioned earlier, I propose that this manuscript should be accepted into Nature Communication after reviewing by the authors. The crystal structure, especially the groove/binding pocket, should be better and more thoroughly compared to other already known CRAL-TRIO structures. Finally, the binding data also needs further verification. The reported dissociation constants are unusually high to draw physiologically relevant conclusions.

1. Panagabko, C. et al. Ligand specificity in the CRAL-TRIO protein family. *Biochemistry* 42, 6467–74 (2003).

helix 4 (CRM-TRO_N)

flanking helices (groove)

mobile gate / lid

a

Kal^bSec14 18 VDAFFRTGSFRNDG-----LKASDVLPI**LKEK**VAFVSGGRDKRGGP**ILTF**PAR-----
 Sec14p 70 QLAKEMFENCEKWRKDYGTDTILQDFH-----YDEKPLIAKFYPQYYHKTDKDRGPVYFEELGAVNLHEMNKVTSE
 Sfh1 66 NASVEMFVETERWREYGAN**TI**IEDYEN**KEA**EDKERIKLAKMYPQYYHHVDKDR**PLY**FEELGGINL**KKM**YKITTE
 Sfh5 73 STIVQNLIDILNWRRE---FN**PL**SCAYKEVHN**TEL**Q-----VGILTFDANGDANK**AVTWN**LY**QGL**VKKKELFQ**N**
 α-TTP 64 DLAWRL**LKN**YYKWRA**EC**P-----EISADLH**PR**SI**IGL**LKAGY**HG**VLRSRDP**TG**SK**VLI**YRIAHWD-----
 SPF 51 -KSEAMLRKHVEFRKQKDIDNIISWQP-----PEVIQ**QY**LS**GG**MCGYDLD**GCP**VWYDI**I**IGPLDAKGL**LF**SASK
 NF1 1581 -----KEEFKAL**KTLS**I**FY**QAGT**SK**AGNP**IF**YV**VAR**-----
 CRALBP 98 RFIRARKFN**VGR**AYEL**LRG**YV**FRL**Q**Y**PELFD**SL**SP**EA**VR**CT**IEAG**Y**PG**VL**SSRD**KY**R**V**VM**LF**NIEN**WQ**-----

Kal^bSec14 66 --SNHDRIRQ**ED**LRKLV**TY**LA**S**VP**SE**DV**CK**RG**FT**VIID**M**RGS-----K**W**DLIK**PL**L**KT**LQ**E**AF**PAE**I**H**VAL**I**IK
 Sec14p 141 ERMLK**N**L**W**WEY**ES**V**V**QY**RL**P**ACS**RA**A**GH**LV**ET**SCT**IM**D**L**K**GIS**ISS**AYS**VM**S**V**RE**AS**Y**IS**Q**NY**PER**M**G**K**FY**I**INA
 Sfh1 143 **KQ**ML**R**N**L**V**K**E**Y**EL**F**AT**Y**RV**P**ACS**R**RA**G**YL**IE**T**SCT**V**L**D**L**K**G**IS**LS**NA**Y**H**V**LS**Y**IK**D**V**AD**IS**Q**NY**PER**M**G**K**F**Y**I**HS
 Sfh5 141 VD**K**F**V**RY**R**IG**L**ME**K**GL**S**LLD---FT**SS**D**NN**Y**MT**Q**V**H**D**Y**K**GV**S**W**VR**MD**SD**IK**NC**SK**T**VI**G**IF**Q**NY**PE**LL**Y**AK**Y**F**V**W**V**
 α-TTP 124 --PK**V**FT**A**Y**D**V**F**RV**S**L**IT**SE**L**IV**Q**E**V**ET**Q**R**NG**IK**A**IF**D**LE**G**W**Q**F**SHA**F**Q**TT**PS**V**A**KK**IA**AV**L**T**D**S**F**PL**K**V**R**GI**H**LI**N**
 SPF 118 QD**LL**RT**K**M**R**E**C**ELL**L**Q**E**CA**H**Q**T**TK**L**G---R**K**V**ET**IT**I**Y**D**CE**G**L**G**L**K**H**L**W**K**PA**V**E**A**Y**G**E**F**L**CM**F**E**EN**Y**P**ET**L**K**R**L**F**V**V**K**
 NF1 1612 --R**F**K**T**Q**ING**D**LL**I**Y**H**V**LL**TL**K**P**Y**AK**---P**Y**E**I**V**D**L**TH**T**G**PS**N**---R**F**K**T**D**F**L**SK**W**F**V**F**PG**F**AY**D**N**V**S**AV**Y**I**Y
 CRALBP 167 -S**Q**E**IT**F**D**E**IL**Q**AY**CF**I**LE**K**L**LE**NE**ET**Q**ING**F**CI**EN**F**K**G**FT**M**Q**QA**AS**L**R**T**SD**L**R**K**M**V**D**M**L**Q**DS**F**PAR**F**KA**I**H**F**I**H**Q

Kal^bSec14 133 PD**N**F**W**Q**K**KT**N**F**G**SS**K**FI**F**ET**S**M**V**S**V**E**G**L**T**K**L**V**D**PS**Q**L**T**EE**F**D**G**SL**D**Y**N**H**E**E**W**I**E**L**R**LS**L**
 Sec14p 218 P**F**G**F**ST**A**F**R**L**K**P**F**LD**P**V**T**V**SK**I**F**IL**G**SS**Y**Q**K**ELL**K**Q**I**PA**EN**LP**V**K**F**G**K**SE**V**DE**S**K**G**LY**L**SD**I**G**P**WR**D**PK
 Sfh1 220 P**F**G**F**ST**M**F**K**M**V**K**P**LD**P**V**T**V**SK**I**F**IL**G**SS**Y**K**K**ELL**K**Q**I**PI**EN**LP**V**K**Y**GG**T**SV**L**H**N**P**N**D**K**F**Y**S**D**I**G**W
 Sfh5 215 P**T**VE**G**W**V**Y**D**L**I**K**F**VE**D**ET**TR**KK**F**V**V**L**T**D**G**SK**L**G**Q**Y**L**K**D**CP**Y**E**G**Y**G**G**K**D**K**KN**L**T**K**Q**N**V**T**N**V**H**P**TE**Y**GL**Y**I**L**Q**K**Q
 α-TTP 199 EP**V**I**F**H**A**V**F**SM**I**K**P**FL**T**E**K**I**K**ER**I**H**M**H**G**N**NY**Q**S**LL**Q**H**F**PD**IL**PLE**Y**GG**E**EF**S**M**E**D**I**C**Q**E**W**T**N**F**I**M**K**S**E**D**Y**L**S**SI
 SPF 194 AP**K**L**F**V**A**Y**N**L**I**K**P**FL**S**E**D**TR**KK**I**M**V**L**G**A**N**W**K**E**V**L**L**K**H**I**SP**D**Q**V**P**V**E**Y**GG**T**M**T**DP**D**GN**P**K**C**K**S**K**I**N**Y**GG**D**I**P**R**K**Y**V**
 NF1 1681 N-C**N**S**W**V**R**E**Y**T**K**Y**H**ER**LL**T**GL**K**G**SK**R**L**V**F**ID**CP**G**KL**A**EH**I**E**H**EQ**KL**PA**AT**L**A**LE**D**L**K**
 CRALBP 243 P**W**Y**F**TT**T**Y**N**V**K**K**P**FL**K**SK**L**LER**V**F**H**GD**DL**SG**F**Y**Q**E**ID**EN**IL**PS**D**FG**G**TL**P**K**Y**D**G**KA**V**A**E**Q**L**F

Panel A – p50RhoGAP vs. α -TTP

Panel B – p50RhoGAP vs. CRALBP

Panel C – p50RhoGAP vs. Sec14p

Reviewer #2:

Remarks to the Author:

The manuscript reports structural and biophysical characterization of lipid binding by the Sec14 superfamily protein Karilin. Karilin is a Rho-GEF protein with diverse functions in the nervous system and significance in disease. The authors determined the structure of the Karilin CRAL_TRIO domain and measured its binding to a panel of lipids. Their crystal structure of the first Rho protein CRAL_TRIO domain surprisingly reveals that the canonical lipid binding pocket is not accessible and that there is a lipid-binding groove on the surface. The authors use nuclear magnetic resonance and chemical shift perturbation mapping to validate lipid binding and the groove as the important binding site. These results therefore have implications for understanding function of Karilin and more broadly other Sec14 family members. The study is technically outstanding, the conclusions are well supported by the data, and the manuscript is for the most part clear and compelling. The article is suitable for publication in Nature Communications after the authors have an opportunity to address the minor concerns and questions below.

1) Although the NMR data well justify the conclusion, the study would be strengthened even more by testing ligand binding to a mutant protein in which one or more of the residues lining the predicted cleft are changed. Experiments could be done in either the NMR binding or flotation assay.

2) While the peaks corresponding to the N-terminal Ex1B front peptide were not assigned and appear disordered, are the authors able to comment on whether any perturbations to those peaks are observed upon ligand binding? Evidence for interaction or structural change in the front peptide upon ligand titration may explain the increased affinity when the front peptide is present in the construct. It would also support the model that front peptides may contribute to ligand discrimination. Another line of evidence supporting the hypothesis that front peptides contribute to ligand binding affinity and specificity would be to measure affinities of the lipid panel for different isoforms produced recombinantly. While not necessary for publication, I do think better supporting this speculation in the discussion would increase the potential impact of the work.

3) The take home message of the structural descriptions at the bottom of page 6 are not clear. Are the authors describing the key stabilizing interactions at the core of the a/b fold or the interactions that connect the a/b fold to other elements. A sentence summarizing the purpose of this description would be helpful, and the authors may choose to make a figure showing the described interactions, although it may not be necessary.

4) Does the structure of the groove and any insights from the docking study explain any of the observed lipid preferences, e.g. lack of affinity for 16:0 or DMPC, higher affinity of FC14 vs. FC12. It would be helpful if the authors used the structure and binding results from the panel of lipids to draw conclusions and make predictions that generalize the type of favored lipids.

5) For the plots of Fraction Bound vs. ligand concentration, e.g. Fig. 4A, is the y-value the average of all the peaks that show significant (more than 2 sigma) CSPs? This is not clear in the methods. Also, the authors could plot the standard deviation of that distribution as error bars in the figure, which would give a clearer indication of the precision across different peaks.

6) The comparison of the Kal(bSec14) with other Sec14 family members nicely summarized the lack of an internal ligand binding pocket. It is less clear whether the groove on the surface of the protein is present in other structures. Even if the specific residues lining the groove and showing CSPs are only conserved in closer relatives to Karilin, as discussed at the end of the results, could the authors comment on how prevalent a groove-like feature is in the other Sec14 sub-family CRAL_TRIO domain proteins that may serve as an additional/alternative binding site?

7) The authors should consider reporting in Supplementary Table 1 some indicator(s) of the quality of the anomalous diffraction data e.g. CCano.

Reviewer #1.

“This work presents novel and valuable insights into the research field of the Kalirin protein.”
“Nevertheless, it is an essential paper to understand the function and mode of action of Kalirins. The elucidation of the first crystal structure of the CRAL-TRIO domain in Kalirin is a significant finding that will positively promote this research field. Furthermore, the experiments were carried out carefully and correctly. The data and its analysis are also meticulous. All experiments are sufficiently documented so that other experimenters can reproduce the experiments without further ado. All in all, this paper is extensive and solid work; the only flaw I see is the wording in some conclusions that should be attenuated.”

We thank the reviewer for his/her enthusiasm for our study. We also appreciate the reviewer’s conclusion that “this manuscript should be accepted into Nature Communication after reviewing by the authors”. We think that revisions made to the text and data obtained from our analysis of two Kal^{bSec14} mutant proteins address the concerns expressed about our conclusions.

Point 1. “In addition, the authors have identified a locus on the surface that forms an elongated groove featuring a dipole-like surface charge that preferentially binds phospholipids with positively charged head groups such as phosphatidylethanolamine and phosphatidylcholines (PE & PC).”

Response: Our lipid binding data show that the surface groove in Kal^{bSec14} specifically binds to lysophospholipids. In the revised manuscript, we make it clear that the dipole-like surface charge identified as a feature of the elongated groove does not confer the ability to bind phosphatidylethanolamine or phosphatidylcholine. Consistent with this conclusion, and as stated in the Discussion, the bar-codes thought to be essential for these head-group interactions in the CRAL_TRIO sub-family proteins are not present in Kalirin.

Point 2. “With all this evidence, the authors must review their structure in a more thorough comparison and re-think their conclusion regarding the binding groove as a novel site. There is strong evidence that the groove is part of the known binding site featuring a novel binding mode.”

Response: We thank the reviewer for assembling two figures and encouraging us to look in more detail at the Kal^{bSec14} groove. To do so, we superimposed the structure of Kal^{bSec14} with those of six CRAL_TRIO domain containing proteins and the BCH domain of p50RhoGAP; we then analyzed the distribution of internal and exposed pockets for each structure (see new Supplementary Fig. 4 and 17). As described in the manuscript, Kal^{bSec14} adopts a closed conformation and does not contain a sizable buried pocket aligned with the canonical Sec14 ligand binding site. The surface groove identified in Kal^{bSec14}, shown by the red arrow in Supplementary Fig. 4a, is completely exposed, with a dipole-like surface charge distribution and is delimited from the gating helix by two helices (H5 and H6).

In contrast, each of the six CRAL_TRIO sub-family proteins, including Sec14p (PDB ID code 1AUA), CARALBP (3HY5), Sfh1 (3B7N), Sfh5 (6W32), NF1 (2D4Q), and α -TTP (3W67), encloses a large internal hydrophobic pocket (indicated by the black arrow) extending from the gating helix to cover most part of the central β sheet, which also serves as the ligand binding site

(Supplementary Fig. 4b). Importantly, none of these six CRAL_TRIO domains possess an exposed surface groove in a location similar to that of Kal^{bSec14}, suggesting that the surface groove serves as a distinct ligand binding site for Kal^{bSec14}.

A CRAL_TRIO_N domain precedes the CRAL_TRIO domain in five of these proteins (Sec14p, CARALBP, Sfh1, Sfh5 and α -TTP). Each of these CRAL_TRIO_N domains is distant from the location of the presumed surface groove but adjacent to the bottom of the gating helix, allowing it to interact with the head group of the bound ligand (Supplementary Fig. 4b). Like Kal^{bSec14}, the Sec14 domain of NF1 lacks a CRAL_TRIO_N domain. Instead of a spectrin-like repeat, the Sec14 domain of NF1 is followed by a PH-like domain; the PH domain does not shield the presumed groove in the NF1 CRAL_TRIO domain (Supplementary Fig. 4b).

In the structure of α -TTP complexed with both α -tocopherol (α -Toc) and phosphatidylinositol 4,5-bisphosphate (PIP2), α -Toc is bound deep in the hydrophobic core of α -TTP, while PIP2 is found in close proximity to the gating helix (Supplementary Fig. 4b). As pointed out by the reviewer, the binding site for α -Toc is right below the site equivalent to the surface groove in Kal^{bSec14}. However, while the Kal^{bSec14} groove is exposed, the α -Toc binding site is entirely sequestered from bulk solvent.

The BCH domain of p50RhoGAP (PDB ID code 7E0W) is distinguished by two exposed pockets, one in a location similar to that of the Kal^{bSec14} surface groove and the other close to the gating helix (Supplementary Fig. 17); the two pockets are separated by the α 4 helix of BCH. A molecule of TEG co-purified with the protein is found in the first pocket, in a straight up pose with one end diving into the hydrophobic core and the other end projecting out of the surface. We agree with the reviewer that part of the TEG binding site in the core is congruent with the α -Toc site in α -TTP. However, α -Toc is completely buried in the core, while TEG is partially exposed to the surface.

Our comparison of the CRAL_TRIO domains of Kal^{bSec14}, BCH and these six proteins, suggests that the ligand binding pocket in BCH represents an intermediate form between the exposed surface cavity in Kal^{bSec14} and the buried pockets in the canonical CRAL_TRIO domains. The data presented in new Supplementary Fig. 4 and 17 are presented in the revised Result and the Discussion, and we have tempered our language in the Discussion to reflect the possibility that the binding site in BCH is in an intermediate form.

Point 3. “Intriguingly, little binding was observed when DMPC was used, featuring two 14-carbon acyl chains instead of one, but no comment on possible reasons for this observation is postulated.”

Response: This important point is addressed in more detail in our response to Point 5. As DMPC has the same head group and the acyl chain length as LPC14:0, the most obvious suggestion is that DMPC, with its two acyl chains, is simply too large to fit into the surface groove of Kal^{bSec14}. However, our docking models (calculated using a flexible ligand docking protocol to allow rotation of any eligible bonds in the ligands) indicate that either lipid can be docked to the open surface groove with different binding modes and ligand conformations; similar docking results have been observed for lipids with longer acyl chains (Supplementary Fig. 14 and data

not shown). It is also possible that when the lipid molecule is embedded in the membrane, the conformation of the lipid chain is restricted, enhancing its binding specificity with Kal^{bSec14}. As discussed in our response to Point 5, we hope to develop the tools needed to explore physiological lipid binding to Kal^{bSec14} at relevant locations in a cellular environment in the future.

Point 4. “This controversial finding is quite puzzling; either the binding affinities are wrong, and another measurement method should be employed to cross-check the values, or the binding is not that specific, hence wise questioning its physiological relevance. The authors do not discuss the meaning of the reported low affinities nor their implications in a physiological context.”

Response: The ~20 μM K_D of Kal^{bSec14} for LPC14:0 is distinctly different from the nanomolar K_D values determined for the interactions of α -TTP/Sec14p/SPF/CRALBP and their ligands using the α -Toc competition assay¹. However, we are confident in the accuracy of our K_D measurements and in their relevance to the physiological role of Kalirin.

Accuracy: When we first started this project, we employed isothermal titration calorimetry (ITC), generally considered the gold standard for measuring binding affinity, to determine the K_D s for Kal^{bSec14}-phosphocholine (FC) detergent interactions. However, we found that the incremental heat changes for each step of the titration were quite low, even when we used rather high concentrations of Kal^{bSec14}, consistent with the occurrence of a low-affinity interaction. The estimated K_D s based on our preliminary ITC data were ~30 and ~500 μM for FC14 and FC12, respectively (data not shown); no ITC signal was observed between Kal^{bSec14} and FC10. Knowing this, we adopted the NMR chemical shift mapping method because it is both a better technique for measuring weak interactions and the only technique allowing localization of the binding site from the same set of the measurements. Nevertheless, the K_D values estimated from our preliminary ITC measurements were quite similar to those obtained by NMR.

For each CSP titration series, we calculated the average K_D using the normalized data from multiple peaks with large CSPs (see the Figure Legend for each K_D plot). As addressed in more detail in our response to Point 5 of Reviewer #2, we have now included the standard deviations for all of the K_D plots. Raw data used to calculate K_D values for each selected peak are included in the Source Data.

Our CSP titration data clearly demonstrated that Kal^{bSec14} specifically recognizes and binds to LPC14:0 since the binding affinity between Kal^{bSec14} and LPC14:0 is about 16-fold higher than that of LPC12:0. In addition, data obtained using FC detergents with hydrocarbon chains of varying lengths and LPC lipids with acyl chains of varying length were in agreement. Where comparisons are possible, data obtained using our flotation assays and NMR titrations are in agreement. Importantly, as addressed in our response to Point 1 of Reviewer #2, two paired mutations targeted to key residues in the surface groove of Kal^{bSec14} reduced its affinity for LPC14:0 by factors of 7- and 13-fold.

Relevance to function: We believe that Kal^{bSec14} exhibits low affinity for its ligands, perhaps for three physiologically relevant reasons. First, the CRAL_TRIO domain in Kalirin is a lipid binding module that likely regulates the subcellular localization of the protein. As such, the

ligand binding site identified in Kal^{bSec14} is exposed on the surface so that it can interact readily with specific lipids embedded in the cellular membrane. In contrast, most of the canonical CRAL_TRIO proteins known to bind their ligands with nM affinities act as lipid transfer proteins, integrating the regulation of lipid metabolism and membrane trafficking^{2,3}. These transfer proteins must be able to recognize a specific lipid with high affinity in order to carry the ligand in its internal ligand binding pocket buried in the protein core to the “acceptor” membrane.

Second, the Sec14 domain in Kalirin does not function as a free agent – it is attached to a much larger protein, which has been shown to interact with a wide variety of other proteins. In neurons, these additional interactions position Kalirin in the post-synaptic density at the post-synaptic membrane, placing its Sec14 domain in the vicinity of the lipid bilayer. A single, high-affinity, Sec14-mediated interaction is unnecessary. In contrast, each of the canonical CRAL_TRIO proteins known to bind their ligand with high affinity is a stand-alone Sec14 domain protein.

Third, work from multiple laboratories using a wide range of systems indicate that Kalirin plays an essential role in the downstream signaling from GPCRs, receptor tyrosine kinase, cell adhesion molecules and ligand gated ion channels⁴⁻⁹. The extensive Ser/Thr and Tyr phosphorylations of Kalirin, along with its multiple protein-protein interaction domains (Sec14, spectrin repeats, PH and SH3) and multiple short linear peptide motifs (PDZ-, SH3, and SH2-binding) classify Kalirin as a signaling hub. It is widely recognized that rapid responses dedicated to diverse cellular needs require low affinity (1-100 μ M), readily reversible interactions amongst multiple proteins; modularity of recognition domains and simultaneous use of multiple suboptimal interactions ensures an overall more specific interaction¹⁰⁻¹⁴.

In the revised manuscript, we introduce the role of Kalirin in signaling pathways in the Introduction and discuss the importance of multiple low affinity protein/peptide and protein/lipid interactions in dynamic systems in the revised Discussion.

Point 5. “Primarily, the meaning of the binding of the proposed ligands (LPC 12:0 and LPC 14:0) is not elucidated enough. The physiological role of these lipids is not discussed. Therefore, I’m surprised by the bold wording in some passages. I don't think that this paper has provided enough insight to make such conclusive statements.”

Response: For the vast majority of Sec14 superfamily members, the identity of their physiologically relevant ligands and their binding affinities remain unknown¹⁵. To the best of our knowledge, this fundamental information is available only for Sec14p (PC or PI), Sfh1 (PtdEtn or PtdIns or PtdCho), Sfh5 (Heme), CRALBP (9-cis-retinal), α -TTP (α -tocopherol and PIP) and SPF (oxidoqualene and squalene); it has been suggested that PtdGro, PtdEtn and PtdCho are the potential physiological ligands for NF1¹⁶.

Our data demonstrate that Kal^{bSec14} binds LPC and LPE but not LPS or LPI, indicating its preference for a positively charged head group; our new mutagenesis data confirms the role of the surface groove in this interaction. We know that Kal^{bSec14} can accommodate a single hydrocarbon (FC14) or acyl chain (LPC14:0) but not DMPC, with its two 14-carbon acyl chains.

While our data suggest a role for phospholipase activation in generating a lysophospholipid that might bind to Kal^{bSec14}, acyl chains with 14 carbons are generally not very prevalent. Whether membrane curvature or membrane disruption associated with vesicular trafficking might expose a limited region of longer acyl chains has not yet been assessed. New methods are being developed to allow the assessment of lipid/protein interactions in a cellular environment with subcellular specificity^{10,17}. A focus on developing these approaches should help us identify the physiologically relevant interactions of lipids with Kalirin. These points have been included in the revised Discussion.

Reviewer #2:

“These results therefore have implications for understanding function of Kalirin and more broadly other Sec14 family members. The study is technically outstanding, the conclusions are well supported by the data, and the manuscript is for the most part clear and compelling.”

We thank the reviewer for his/her critical and constructive reviews of our manuscript. It helped us clarify several important issues that reinforce our conclusion and made the revision a better paper. We also appreciate the reviewer’s conclusion that “the article is suitable for publication in Nature Communication after the authors have an opportunity to address the minor concerns and questions below”. We hope that the revised manuscript meets with approval.

Point 1. “Although the NMR data well justify the conclusion, the study would be strengthened even more by testing ligand binding to a mutant protein in which one or more of the residues lining the predicted cleft are changed. Experiments could be done in either the NMR binding or flotation assay.”

Response: We completely agree with the reviewer on this point. To this end, we have designed and generated two double Kal^{bSec14} mutants (D69A/R70A and S105A/K106A) that carry alanine substitutions at two loop regions surrounding the surface groove. These four amino acid residues were selected not only because of their large CSP signals during NMR titrations, but also for their locations in the loops, minimizing potential disturbance of protein folding. Size exclusion chromatography and NMR spectra of the mutant proteins confirm that both proteins remain well folded in solution. NMR titrations of the mutants with LPC14:0 show that Kal^{bSec14}D69A/R70A and Kal^{bSec14}S105A/K106A have K_D s of $129 \pm 7 \mu\text{M}$ and $244 \pm 21 \mu\text{M}$, respectively (new Fig. 7 and Supplementary Fig. 13), as compared to the K_D of $18.9 \pm 2.9 \mu\text{M}$ for the wild-type protein. When tested qualitatively with LPC14:0 in the flotation assay, neither mutants exhibited binding above the background levels (data not shown). These results further reinforce our finding that the surface groove of Kal^{bSec14} serves as the ligand binding site.

Point 2. “While the peaks corresponding to the N-terminal Ex1B front peptide were not assigned and appear disordered, are the authors able to comment on whether any perturbations to those peaks are observed upon ligand binding? Evidence for interaction or structural change in the front peptide upon ligand titration may explain the increased affinity when the front peptide is present in the construct. It would also support the model that front peptides may contribute to ligand discrimination. Another line of evidence supporting the hypothesis that front peptides contribute to ligand binding affinity and specificity would be to measure affinities of the lipid

panel for different isoforms produced recombinantly. While not necessary for publication, I do think better supporting this speculation in the discussion would increase the potential impact of the work.”

Response: Clarification of the role of the front peptides in different Kalirin isoforms is important for understanding its ligand specificity and affinity. The N-terminal Ex1B front peptide in Kal^{bSec14} consists of 24 amino acids including two proline residues, and is negatively charged at physiologic pH (isoelectric point of 3.5; new Supplementary Fig. 15a). When we examined the superimposed HSQC spectra of Kal^{bSec14} and Kal^{Sec14}, we identified 22 unique peaks present only in the spectrum of Kal^{bSec14} (see the revised Supplementary Fig. 5c). The majority of these peaks are clustered in the middle part of the spectrum corresponding to the N-terminal disordered region, while the three presumed glycine peaks are found on the upper part of the spectrum. None of the 22 unassigned peaks were shifted upon titration with increasing amounts of LPC14:0 (Supplementary Fig. 10a), indicating that the Ex1B front peptide does not interact directly with this lipid. We also did not identify any new peak during the course of the titration.

Chemical shift mapping of the superimposed HSQC spectra of Kal^{bSec14} and Kal^{Sec14} revealed that the residues displaying significant CSPs ($> 2\sigma$) are located in four regions, the assigned N-terminal loop (F27 and R28), strand S1 and the following S1–S2 loop (V48–G51), helix H4 (E89 and D90) and the H6–S4 loop (A123) (new Supplementary Fig. 15b). Although these structural segments are not part of the surface groove, they surround its rear portion (Supplementary Fig. 15c, d). Interestingly, a network of basic residues located adjacent to these CSP-sensitive regions, including Arg52, Lys54, Arg55, Lys93, Arg94 and His126, forms a large positively charged patch on the surface that could be readily accessed by the Ex1B peptide (Supplementary Fig. 15d and e). It is thus possible that the Ex1B front peptide affects lipid binding indirectly by interacting transiently with the core domain of the protein so as to change protein dynamics to populate a conformation optimal for ligand binding. We should point out that the observed chemical shift differences between the spectra of Kal^{bSec14} and Kal^{Sec14} do not appear to be caused by a global conformational change in the protein; we were therefore able to faithfully transfer the assignment from Kal^{Sec14} to Kal^{bSec14}. The conclusions drawn from these new analyses have been added to a new section in the revised Results.

Based on the limited data available, transcripts containing Ex1B or Ex1C predominate in human, mouse and rat¹⁸. We agree with the reviewer that it is important to understand why Kal^{bSec14} binds LPC while Kal^{cSec14} recognizes phosphoinositides. The amphipathic helix encoded by Ex1C¹⁹ limits the solubility of Kal^{cSec14} and cKal7, precluding measurement of their affinity for specific lipids (Supplementary Fig. 15a). Despite testing a variety of expression systems (including insect cells and mammalian cells), varying extraction conditions and detergents and expressing a number of different Kal^{cSec14}-containing constructs, we were unable to produce proteins that could be used for structural and biophysical studies. Kalirin transcripts that include Ex1D are found only in rat and mouse⁵ (Supplementary Fig. 15a). The Ex1A front peptide consists of only five amino acid residues and is expected to share the ligand preferences of Kal^{Sec14} (Supplementary Fig. 15a). Perhaps most importantly, the lipid binding specificities of Kal^{bSec14} and Kal^{cSec14} need to be assessed in intact cells under physiological conditions.

Point 3. “The take home message of the structural descriptions at the bottom of page 6 are not

clear. Are the authors describing the key stabilizing interactions at the core of the a/b fold or the interactions that connect the a/b fold to other elements. A sentence summarizing the purpose of this description would be helpful, and the authors may choose to make a figure showing the described interactions, although it may not be necessary.”

Response: It is now clear that the $\alpha\beta\alpha$ sandwich fold first identified in yeast Sec14p is common to all members of the Sec14 protein superfamily. In other stand-alone Sec14 domains, the preceding CRAL_TRIO_N domain forms a key part of the ligand binding pocket. CRAL_TRIO domains that are not preceded by a CRAL_TRIO_N domain occur in NF1, p50RhoGAP, Kalirin and other SecSR proteins. In NF1, the PH domain that immediately follows its CRAL_TRIO domain stabilizes its $\alpha\beta\alpha$ fold²⁰. In p50RhoGAP, the intertwined dimer of the BCH domain stabilizes each monomer's $\alpha\beta\alpha$ fold²¹.

In Kalirin, the Kal^{bSec14} construct was selected for this study because we were unable to obtain diffraction-quality crystals of the CRAL_TRIO domain in the absence of the A helix of the first SR. As revealed by the structure of Kal^{bSec14}, the SR helix interacts with the CRAL_TRIO domain to stabilize its $\alpha\beta\alpha$ fold. As suggested by the reviewer, we have now summarized this point at the end of the paragraph.

Point 4. “Does the structure of the groove and any insights from the docking study explain any of the observed lipid preferences, e.g. lack of affinity for 16:0 or DMPC, higher affinity of FC14 vs. FC12. It would be helpful if the authors used the structure and binding results from the panel of lipids to draw conclusions and make predictions that generalize the type of favored lipids.”

Response: Reviewer #1 (point 3) posed a similar question; please see our response to this important question.

Point 5. “For the plots of Fraction Bound vs. ligand concentration, e.g. Fig. 4A, is the y-value the average of all the peaks that show significant (more than 2 sigma) CSPs? This is not clear in the methods. Also, the authors could plot the standard deviation of that distribution as error bars in the figure, which would give a clearer indication of the precision across different peaks.”

Response: We thank the reviewer for requesting clarification. For each titration series, we first ranked all of observed peaks based on their CSPs upon titration at the highest concentration of the ligands. We then selected the top-ranked peaks with large as well as well-resolved and non-overlapped CSPs for each titration point. These peaks always contained most of the 2σ CSPs and some of the 1σ CSPs. The y-values shown in the K_D plots are the normalized and averaged CSPs for all the selected peaks. Standard deviations are now included in the plots. We now explain our approach in the Methods and in the Figure Legends. The raw data used to calculate K_D values for each selected peak are included in the Source Data being submitted.

Point 6. “The comparison of the Kal(bSec14) with other Sec14 family members nicely summarized the lack of an internal ligand binding pocket. It is less clear whether the groove on the surface of the protein is present in other structures. Even if the specific residues lining the groove and showing CSPs are only conserved in closer relatives to Karilin, as discussed at the end of the results, could the authors comment on how prevalent a groove-like feature is in the

other Sec14 sub-family CRAL_TRIO domain proteins that may serve as an additional/alternative binding site?”

Response: This is an important point. As addressed in our response to Point 2 of Reviewer #1, we accessed the distribution of internal and exposed pockets in six CRAL_TRIO subfamily proteins. None has an exposed groove similar to the Kal^{bSec14} surface groove (new Supplementary Fig. 4b). We expanded our discussion of this new analysis in the revised Results and the Discussion.

Point 7. “The authors should consider reporting in Supplementary Table 1 some indicator(s) of the quality of the anomalous diffraction data e.g. CC_{ano}.”

Response: We now report three parameters for the anomalous diffraction data, Anomalous completeness, Anomalous multiplicity and CC_{ano}, in Supplementary Table 1.

References

1. Panagabko, C. et al. Ligand specificity in the CRAL-TRIO protein family. *Biochemistry* **42**, 6467-74 (2003).
2. Bankaitis, V.A., Mousley, C.J. & Schaaf, G. The Sec14 superfamily and mechanisms for crosstalk between lipid metabolism and lipid signaling. *Trends Biochem. Sci.* **35**, 150-60 (2010).
3. Ile, K.E., Schaaf, G. & Bankaitis, V.A. Phosphatidylinositol transfer proteins and cellular nanoreactors for lipid signaling. *Nat. Chem. Biol.* **2**, 576-83 (2006).
4. Schiller, M.R. et al. Regulation of RhoGEF activity by intramolecular and intermolecular SH3 domain interactions. *J. Biol. Chem.* **281**, 18774-86 (2006).
5. Miller, M.B. et al. Alternate promoter usage generates two subpopulations of the neuronal RhoGEF Kalirin-7. *J. Neurochem.* **140**, 889-902 (2017).
6. Penzes, P. et al. The neuronal Rho-GEF Kalirin-7 interacts with PDZ domain-containing proteins and regulates dendritic morphogenesis. *Neuron* **29**, 229-42 (2001).
7. Kiraly, D.D., Eipper-Mains, J.E., Mains, R.E. & Eipper, B.A. Synaptic plasticity, a symphony in GEF. *ACS Chem. Neurosci.* **1**, 348-365 (2010).
8. Bircher, J.E. & Koleske, A.J. Trio family proteins as regulators of cell migration and morphogenesis in development and disease - mechanisms and cellular contexts. *J. Cell Sci.* **134**(2021).
9. Parnell, E. et al. KALRN: A central regulator of synaptic function and synaptopathies. *Gene* **768**, 145306 (2021).
10. Bagheri, Y., Ali, A.A. & You, M. Current methods for detecting cell membrane transient interactions. *Front. Chem.* **8**, 603259 (2020).
11. Gibson, T.J. Cell regulation: determined to signal discrete cooperation. *Trends Biochem. Sci.* **34**, 471-82 (2009).
12. Stein, A., Pache, R.A., Bernado, P., Pons, M. & Aloy, P. Dynamic interactions of proteins in complex networks: a more structured view. *FEBS J.* **276**, 5390-405 (2009).

13. Mayer, B.J. Perspective: Dynamics of receptor tyrosine kinase signaling complexes. *FEBS Lett.* **586**, 2575-9 (2012).
14. Perkins, J.R., Diboun, I., Dessailly, B.H., Lees, J.G. & Orengo, C. Transient protein-protein interactions: structural, functional, and network properties. *Structure* **18**, 1233-43 (2010).
15. Saito, K., Tautz, L. & Mustelin, T. The lipid-binding SEC14 domain. *Biochim. Biophys. Acta* **1771**, 719-26 (2007).
16. Welti, S., Fraterman, S., D'Angelo, I., Wilm, M. & Scheffzek, K. The Sec14 homology module of neurofibromin binds cellular glycerophospholipids: mass spectrometry and structure of a lipid complex. *J. Mol. Biol.* **366**, 551-62 (2007).
17. De Keersmaecker, H. et al. Mapping transient protein interactions at the nanoscale in living mammalian cells. *ACS Nano.* **12**, 9842-9854 (2018).
18. Mains, R.E., Kiraly, D.D., Eipper-Mains, J.E., Ma, X.M. & Eipper, B.A. *Kalrn* promoter usage and isoform expression respond to chronic cocaine exposure. *BMC Neurosci.* **12**, 20 (2011).
19. Miller, M.B., Vishwanatha, K.S., Mains, R.E. & Eipper, B.A. An N-terminal amphipathic helix binds phosphoinositides and enhances Kalirin Sec14 domain-mediated membrane interactions. *J. Biol. Chem.* **290**, 13541-55 (2015).
20. D'Angelo, I., Welti, S., Bonneau, F. & Scheffzek, K. A novel bipartite phospholipid-binding module in the neurofibromatosis type 1 protein. *EMBO Rep.* **7**, 174-9 (2006).
21. Chichili, V.P.R. et al. Structural basis for p50RhoGAP BCH domain-mediated regulation of Rho inactivation. *Proc. Natl. Acad. Sci. U. S. A.* **118**(2021).

Reviewers' Comments:

Reviewer #1:

Remarks to the Author:

I thank the authors for their extensive replies to my comments from the first review. I appreciate that the authors have systematically addressed all the relevant points I brought up for revision. Please, find my comments on each reply addressing every single point.

Point 1

The authors have investigated the dipole-like surface of the groove and were able to determine that the electrostatic surface potential is not a forceful feature for the recognition of the lipid head group. They correctly discussed that in Sec14p homologs, the CRAL-TRIO_N domain is crucial for recognizing lipid head groups. As they pointed out, the fact that the CRAL-TRIO_N domain is missing in Kalirins indicates a limited ability of these proteins to identify the lipid head groups. Strikingly, they describe that the front peptides may modulate the binding preference to lipids with specific head-group partially mimicking the function of the CRAL-TRIO_N domain.

Point 2

The authors have thoroughly analyzed the three-dimensional structure of KalbSec14 and compared it to other available structures of CRAL-TRIO proteins. They acknowledge that the surface cavity of KalbSec14, the binding pocket in BCH, and the canonical binding pocket in CRAL-TRIO proteins may represent intermediate states. I appreciate their inclusion of supplementary figures 4 and 17, where they show and discuss the nature of the surface binding cavity in detail. They identified and highlighted the helices and beta-sheets that contribute to the binding of the ligands in all illustrated CRAL-TRIO proteins. It becomes apparent that the binding is mediated by topologically equivalent secondary structure elements, independent of the binding mode - be it a surface groove or an internal binding pocket resulting in alternate binding sites, as mentioned by the authors in the revised discussion.

Point 3

The authors have given some thought to the strikingly different binding affinities of DMPC and LPC 14:0 to KalbSec14, respectively. Possible explanations, based on logic, were eloquently formulated. As in point 5, the authors discuss the need for further research into Kalirin and its ligands to elucidate complex physiological questions.

Point 4

The authors have addressed my comments systematically regarding the reported low binding affinities and the physiological relevance. There is no doubt that the methods employed for determining the dissociation constants are state-of-the-art. Isothermal titration calorimetry (ITC) and chemical shift perturbation (CSP) are sophisticated methods that produce reliable data. This fact makes the data presented all the more plausible. The argumentation for the physiological relevance of the binding despite low affinities determined in this paper for the Kalirin proteins in a dynamic signaling pathway setting is eloquently discussed.

Point 5

The authors commented on the physiological relevance of the finding in this paper. As they correctly noted, many ligands of proteins in the Sec14 superfamily are unknown, and their physiological connection is still unclear. Fortunately, the authors have thought about how they may explore the physiological relevance of the described lipids and Kalirin in the future. They have acknowledged the need for further investigation and paid tribute to this fact by moderating some conclusions in this early paper.

The authors have acknowledged my comments and addressed them in interesting and constructive replies. They have conducted a thorough and diligent review of their manuscript, resulting in an

excellent manuscript featuring solid data and eloquent conclusions. Therefore, I recommend accepting the reviewed manuscript without further ado in Nature Communication.

Reviewer #2:

Remarks to the Author:

The authors have carefully considered reviewer concerns, and they have added additional experiments and analysis. They also offered additional clarifying explanations to conclusions that were previously not sufficiently clear. For all these reasons, the manuscript is considerably improved and suitable for publication. The work is interesting, technically sound, and conclusions are fairly drawn from the data. The authors should be congratulated on an excellent study that will advance the field.